# CAN LLMS MAINTAIN FUNDAMENTAL ABILITIES UNDER KV CACHE COMPRESSION?

## ABSTRACT

This paper investigates an underexplored challenge in large language models (LLMs): the impact of KV cache compression methods on LLMs' fundamental capabilities. Although existing methods achieve impressive compression ratios on long-context benchmarks, their effects on core model capabilities remain understudied. We present a comprehensive benchmark KVFundaBench to systematically evaluate the effects of KV cache compression across diverse fundamental LLM capabilities, spanning world knowledge, commonsense reasoning, arithmetic reasoning, code generation, safety, and long-context understanding and generation.Our analysis reveals serval key findings: (1) *Task-Dependent Degradation*; (2) *Model-Type Robustness* (3) *Prompt Length Vulnerability*; (4) *Chunk-Level Superiority*; (5) *Prompt-Gain Sensitivity*; (6) *Long-Context Generation Sensitivity*. Based on our analysis of attention patterns and cross-task compression performance, we propose ShotKV, a novel compression approach that distinctly handles prefill and decoding phases while maintaining shot-level semantic coherence. Empirical results show that ShotKV achieves $9\%$-$18\%$ performance improvements on long-context generation tasks under aggressive compression ratios.

## 1 INTRODUCTION

The evolution of Large Language Models (LLMs) to process large documents for tasks such as answering and summarizing questions (Raffel et al., 2020; Brown et al., 2020; Chowdhery et al., 2022; Tay et al., 2022; Touvron et al., 2023a;b), spurred by breakthroughs in system architectures (Dao et al., 2022; Dao, 2024; Jacobs et al., 2023; Xiao et al., 2024) and model design (Chen et al., 2023a; Xiong et al., 2024; Chen et al., 2023b; Peng et al., 2024), has significantly increased GPU memory demands during inference (AI21, 2024; X.AI, 2024; Reid et al., 2024; Anthropic, 2024; DeepSeek-AI, 2024; Liu et al., 2024a), making the development of efficient key value (KV) cache compression strategies a critical focus for LLM deployment and optimization.

To address this, numerous studies have proposed selective token retention strategies (Xiao et al., 2024; Zhang et al., 2023; Li et al., 2024b; Ge et al., 2023; Cai et al., 2024; Fu et al., 2024; Yang et al., 2024; Adnan et al., 2024; Liu et al., 2024e; Tang et al., 2024), with pioneering works such as H2O (Zhang et al., 2023) and SnapKV (Li et al., 2024b) showing that retaining approximately 50% of KV cache entries can balance model performance with significant memory savings. However, current research primarily evaluates these methods in *retrieval-based* long-context scenarios such as LongBench Bai et al. (2023; 2025) and Need-In-A-Haystack (NIAH) Kamradt (2023). This narrow focus overlooks *reasoning-intensive* long-context scenarios, such as many-shot in-context learning (ICL) (Agarwal et al., 2024), where the context length is driven by extensive examples and the challenge lies not merely in retrieving specific information ("needle in a haystack"), but in **maintaining reasoning chains** across extended generation sequences (e.g., 4k+ tokens). In these settings, the pressure on the KV cache comes from the necessity to preserve the semantic coherence required for multi-step deduction. Consequently, the impact of compression on a spectrum of fundamental LLM capabilities—such as *arithmetic reasoning*, *world knowledge*, *commonsense reasoning*, and *safety*—remains largely unexplored, particularly concerning their distinct attention patterns. To this end, we introduce **KVFundaBench**, a benchmark designed to systematically assess the effects of KV cache compression across these diverse fundamental capabilities and their underlying attention dynamics. The benchmark includes 5 categories of tasks: *world knowledge*, *commonsense reasoning*, *arithmetic reasoning*, *code generation*, and *safety*. Our comprehensive

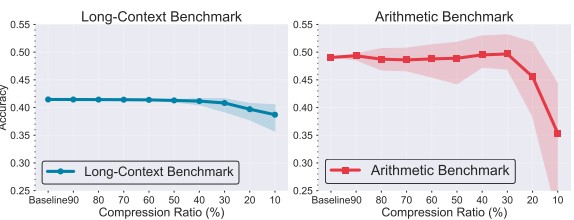 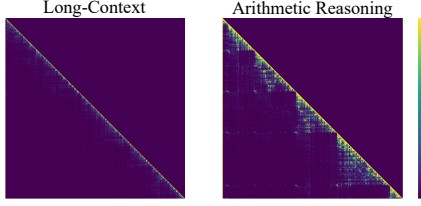

(a) KV cache compression methods on long-context and arithmetic benchmarks.

(b) Attention heatmap on long-context and arithmetic benchmarks.

Figure 1: KV cache compression methods on long-context and arithmetic benchmarks. (a) Arithmetic benchmark shows more performance degradation than long-context benchmark. (b) Long-Context benchmark shows more sparsity in attention heatmap.

evaluations using KVFundaBench reveal several critical findings: we observe, as shown in Figure 1, that arithmetic reasoning tasks suffer significantly higher performance degradation under compression compared to long-context tasks, and that attention patterns in long-context scenarios exhibit notably higher sparsity. These initial results suggest that existing evaluation frameworks, which focus predominantly on long-context performance, may not adequately capture the full impact spectrum. Our KVFundaBench reveals several key findings: (1) *Task-Dependent Degradation*: Performance degradation is highly task-dependent, with arithmetic reasoning tasks showing particular sensitivity to aggressive compression; (2) *Model-Type Robustness*: Multi-step reasoning LLMs demonstrate higher compression robustness compared to instruction-tuned models; (3) *Prompt Length Vulnerability*: Shorter prompts are more vulnerable to compression effects; (4) *Chunk-Level Superiority*: Chunk-level compression strategies show superior performance on complex long-context reasoning tasks; (5) *Prompt-Gain Sensitivity*: Tasks with larger prompt-based performance gains exhibit higher compression sensitivity; and (6) *Long-Context Generation Sensitivity*: Long-context generation tasks are particularly sensitive to compression. These findings provide valuable insights into the relationship between compression methods and model capabilities, motivating our development of **ShotKV**, which is a new KV cache compression method with separate compression methods for prefill and decoding phases.

We hope our work can provide the research community with insightful perspectives on the impact of KV cache compression on LLMs. Our main contributions are summarized as follows:

- Introduce **KVFundaBench** to systematically evaluate the effects of KV cache compression across diverse fundamental LLM capabilities, we demonstrate that task-specific sensitivity to compression varies significantly, with performance degradation ranging from 1% to 40%.
- Our systematic investigation reveals multiple critical factors influencing compression sensitivity, including model training dynamics, prompt length characteristics, task-specific requirements, long-context reasoning, and long-context generation capabilities.
- We introduce **ShotKV**, an innovative compression framework that distinctively manages the prefill and decoding phases while maintaining the semantic integrity of the shot level.

## 2 PRELIMINARY

In this section, we provide comprehensive preliminaries of KV cache compression and LLM evaluation.

**Key-Value Cache in LLMs** With the increasing long-context capabilities of LLMs, the Key-Value (KV) cache has become crucial for improving inference efficiency. During LLM inference, the KV cache stores intermediate computation results to avoid redundant calculations. For a given input sequence $x = (x_1, x_2, ..., x_n)$, each transformer layer $l$ maintains its key cache $K^l = (k_1^l, k_2^l, ..., k_n^l)$ and value cache $V^l = (v_1^l, v_2^l, ..., v_n^l)$, where $k_i^l, v_i^l \in \mathbb{R}^d$ represent the key and value vectors for token $x_i$ at layer $l$.

**KV Cache Compression** KV cache compression aims to reduce memory usage by selectively storing or merging cached vectors. A compression operation can be denoted as $C(K, V) = (K', V')$, where $K'$ and $V'$ are compressed caches with size $m < n$, where $C$ is the compression method, $m$ is the number of retained tokens, and $n$ is the original number of tokens. The core idea is token

Table 1: Hyperparameters for Different Observations

| Benchmarks | Obs 1 | Obs 2 | Obs 3 | Obs 4 | Obs 5 | Obs 6 | |
|---|---|---|---|---|---|---|---|
| | Number of Shots | | | | | $K$ | $T$ |
| MMLU Hendrycks et al. (2020) | 5 | 5 | - | - | 0,5 | - | - |
| CommonsenseQA Talmor et al. (2019) | 4 | 4 | - | - | - | - | - |
| GSM8K Cobbe et al. (2021) | 8 | 8 | 1-8 | 50 | 0,8 | - | - |
| HumanEval Chen et al. (2021) | 8 | 8 | - | - | - | - | - |
| JailBreakV Luo et al. (2024) | 8 | 8 | - | - | - | - | - |
| LongGenBench-GSM8K Liu et al. (2024d) | - | - | - | - | - | 35 | 20 |

selection - identifying and retaining important tokens based on attention patterns or other metrics while discarding less important ones. The compression ratio $r = m/n$ indicates how aggressively the cache is compressed, where a smaller ratio means more aggressive compression.

**Evaluation Protocol** To thoroughly evaluate the impact of KV cache compression on LLMs' capabilities, we assess five benchmark categories: world knowledge, commonsense reasoning, arithmetic reasoning, code generation, and safety.

For each task category and compression method $C$, we calculate the relative performance change as follows:

$$\Delta P = \frac{P_C - P_{\text{base}}}{P_{\text{base}}} \tag{1}$$

where $P_C$ and $P_{\text{base}}$ represent the performance scores with and without compression, respectively.

## 3 BENCHMARK DESIGN

### 3.1 BENCHMARK SETUPS

In this section, we will introduce the KVFundaBench setups, including the datasets, models, and evaluation environment.

**Datasets** To evaluate the performance of KV cache compression on LLMs' overarching capabilities, we assess five benchmark categories: *World Knowledge (WK)* using MMLU (Hendrycks et al., 2020), measured by accuracy; *CommonSense Reasoning (CSR)* using CommonsenseQA (Talmor et al., 2019), evaluated through multiple-choice accuracy; *Arithmetic Reasoning (AR)* using GSM8K (Cobbe et al., 2021), assessed by solve rate; *Code Generation (CG)* using HumanEval (Chen et al., 2021), measured by pass@1 rate on test cases; and *Safety (SA)* using JailBreakV (Luo et al., 2024), evaluated by attack success rate. Furthermore, we test the performance of KV cache compression on LongGenBench (Liu et al., 2024d), a *long-context generation (LG)* benchmark. Detailed statistics for all datasets are provided in Section E.1.

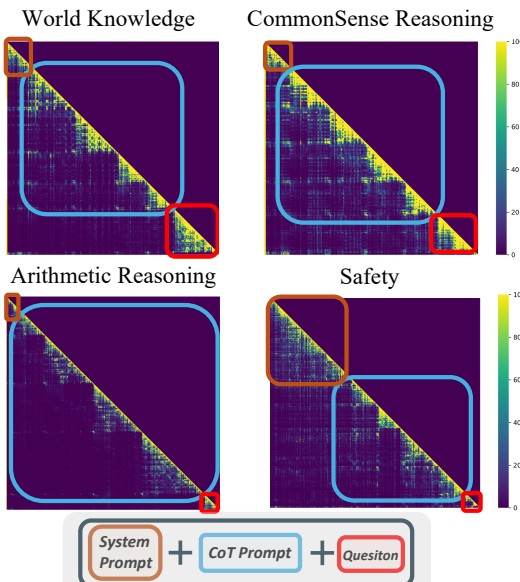

Figure 2: Attention heatmap on different tasks.

**Models** We conduct experiments on a series of LLMs, including LLaMA-3.1-8B, LLaMA-3.1-8B-Instruct (Dubey et al., 2024), Mistral-7B-Instruct (Jiang et al., 2023a), and multi-step reasoning LLM DeepSeek-R1-Distill-Llama-8B (Guo et al., 2025).

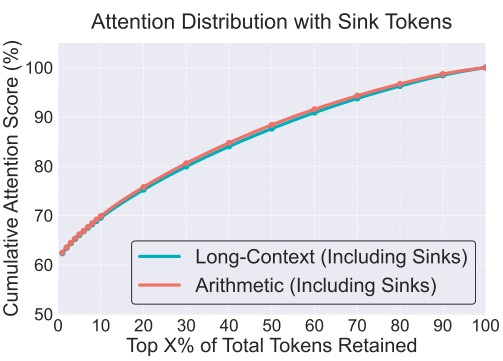 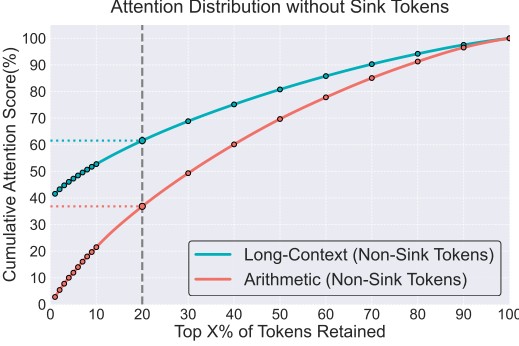

(a) Attention Distribution with Sink Tokens     (b) Attention Distribution without Sink Tokens

Figure 3: Cumulative attention score distribution for Long-Context and Arithmetic. (a) Overall distribution including initial sink tokens, showing high initial concentration. (b) Distribution without sink tokens (first 4 tokens removed), revealing that Arithmetic's non-sink attention is more diffuse compared to Long-Context's.

**KV Cache Compression Methods** To thoroughly investigate the potential impact on KV cache compression methods, we select the following methods: StreamingLLM Xiao et al. (2024), SnapKV Li et al. (2024b), H2O Zhang et al. (2023), PyramidKV Cai et al. (2024), PyramidInfer Yang et al. (2024), and ChunkKV Liu et al. (2025).

**Hyperparameters** The hyper-parameters for different observations are shown in Table 1. The temperature for the experiments are set to 0 for ensuring the deterministic results.

## 3.2 ATTENTION PATTERN ANALYSIS ON KVFUNDABENCH

To better understand the task-specific sensitivity, we analyze the Cumulative Distribution Function (CDF) of attention scores, as shown in Figure Figure 2. Based on the slope and concentration of the CDF curves, we categorize task attention patterns into two distinct types:

**Universal Patterns (WK/CSR):** As observed in World Knowledge and Commonsense Reasoning, the attention distribution is relatively uniform (after excluding sink tokens). The CDF curve rises smoothly, indicating that the model aggregates information from a broad range of context tokens. This "bag-of-words" style attention is robust to compression because losing a small fraction of tokens does not critically disrupt the overall semantic representation.

**Specialized Patterns (AR):** In Arithmetic Reasoning tasks, the attention pattern is highly sparse and specialized. The CDF curve for non-sink tokens is significantly flatter (Figure 3b), implying that the model concentrates its attention mass on a very small, specific set of tokens—likely the intermediate steps crucial for the reasoning path. We term this a "Specialized" pattern. Unlike retrieval tasks, these tokens act as "bridges" in a reasoning chain; if compression algorithms (like H2O or SnapKV) mistakenly discard these key tokens, the entire **Chain-of-Thought (CoT) is broken**, leading to the severe performance degradation we observed.

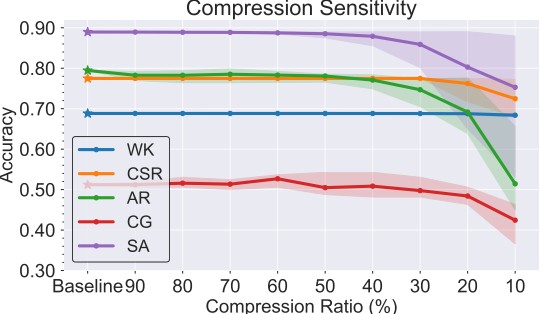

(a) Sensitivity Analysis of Different Benchmark Categories to KV Cache Compression

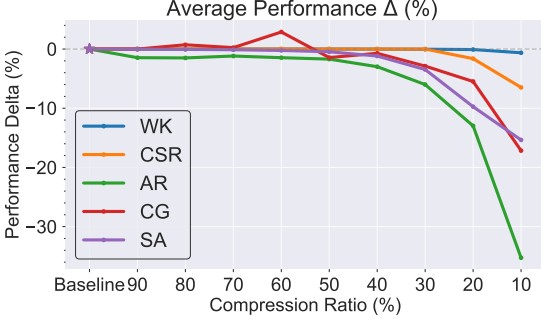

(b) Performance Delta Lines with Baseline

Figure 4: Sensitivity Analysis of Different Benchmark Categories to KV Cache Compression. The performance delta lines are calculated by Equation (1).

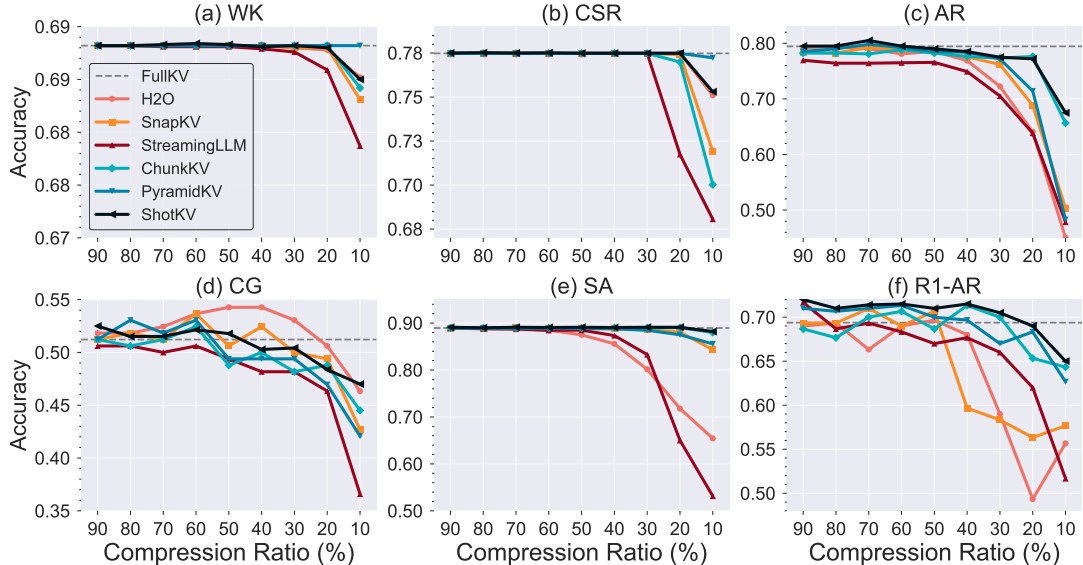

Figure 5: Performance Comparison of KV Cache Compression Methods on KVFundaBench. Results for R1-AR (f) were obtained using the DeepSeek-R1-Distill-Llama-8B model. ShotKV is our proposed method; details can be found in Section 4.

To further investigate the attention dynamics that might explain the task-specific sensitivities to KV cache compression, we analyzed the cumulative attention score distributions, as illustrated in Figure 3. Figure 3(a) depicts the overall attention distribution, which includes the initial sink tokens Xiao et al. (2024). In this view, both long-context and arithmetic tasks demonstrate a very similar pattern: a steep initial rise where the top 1% of tokens capture over 60% of the total attention mass. This highlights the predominant role of sink tokens in attracting attention, regardless of the specific task.

However, a more distinct pattern emerges when these initial sink tokens (specifically, the first four tokens) are excluded from the analysis, as shown in Figure 3(b). Within the remaining non-sink tokens, the attention distribution for arithmetic tasks becomes notably more diffuse, with a slower accumulation of attention mass. For instance, the top 20% of non-sink tokens in arithmetic cover only about 37% of the attention within their own non-sink group. In contrast, long-context's non-sink tokens exhibit a relatively more concentrated attention profile, where the top 20% of its non-sink tokens capture approximately 61.5% of the attention within their non-sink set. This divergence suggests that while sink tokens provide a common, strong attentional anchor, the subsequent distribution of attention across task-relevant (non-sink) tokens varies. The more diffuse attention in arithmetic's non-sink tokens implies a reliance on a broader set of contextual cues for its structured reasoning, potentially making it more vulnerable when compression begins to impact these non-sink tokens.

These detailed analyses of attention distributions ( Figure 2 and Figure 3) reveal that LLMs engage different contextual information and attention strategies when performing long-context tasks versus tasks requiring fundamental abilities such as arithmetic reasoning. This highlights the necessity of evaluating KV cache compression beyond long-range dependencies to specifically assess its impact on diverse fundamental capabilities, owing to their distinct attentional mechanisms.

## 3.3 Results and Analysis

In this section, we present the results and an analysis of the experiments. For detailed results, see Section C.1.

**Evaluation Environment** We use the lm-evaluation-harness (Gao et al., 2023) library to load the models and evaluate the performance. The evaluation environment is a NVIDIA A40 GPU server.

*Observation 1.* **Task-Dependent Degradation**: KV cache compression methods show task-dependent performance degradation, WK and CSR are more robust to KV cache compression. As demonstrated in Figure 4, all tasks maintain stable performance at compression ratios above 40%, but exhibit distinct degradation patterns below this threshold. *Arithmetic reasoning*, *code genera-*

*tion*, and *safety* tasks demonstrate the highest compression sensitivity, characterized by precipitous performance declines. Figure 5 illustrates the detailed performance impact of various KV cache compression methods across different tasks. This degradation is most pronounced in *arithmetic reasoning* (c), where performance deteriorates significantly below the compression ratio of 20%, with precision dropping from approximately 0.75 to below 0.5. Among the evaluated methods, ChunkKV Liu et al. (2025) and PyramidKV Cai et al. (2024) consistently demonstrate superior stability in most tasks, while StreamingLLM Xiao et al. (2024) exhibits increased sensitivity to aggressive compression. Additionally, *R1-Arithmetic reasoning* (f) indicates that reasoning LLMs demonstrate enhanced robustness to KV cache compression. Highlighting *World Knowledge* and *Common Sense Reasoning* as the most robust tasks, indicating that these tasks are less sensitive to KV cache compression.

*Observation 2.* **Model-Type Robustness**: Multi-step reasoning LLMs are more robust to KV cache compression. Figure 7 presents a comparative analysis of LLaMA-3.1-8B across its base (w/o instruct tuned), instruct-tuned, and DeepSeek-R1 distilled variants, illustrating their averaged performance in five compression methods with confidence intervals. Although all three variants exhibit performance degradation at low compression ratios, their degradation trajectories differ significantly. The R1 distilled model demonstrates superior stability, maintaining performance around 0.60 even at a compression ratio 10%. Although the instruct-tuned model achieves a higher initial accuracy (0.8), it exhibits heightened compression sensitivity, with performance deterioration beginning at 30% compression ratio and declining sharply to approximately 0.5 at 10% ratio. These findings suggest that while multi-step reasoning LLMs demonstrate enhanced robustness to KV cache compression, and instruct-tuning improves overall model performance, the latter may inadvertently increase model vulnerability to aggressive compression, particularly at compression ratios below 30%.

*Observation 3.* **Prompt Length Vulnerability**: Shorter prompts are more vulnerable to KV cache compression. As illustrated in Figure 8, the effect of KV cache compression is markedly different with varying prompt lengths (shot numbers). Scenarios with fewer shots (for example, one-shot and two-shot) demonstrate heightened sensitivity to compression; their performance degrades more precipitously below a compression ratio of 30% compared to scenarios with a greater number of shots (e.g., 4-8 shots). For example, in 1-shot settings, performance decreases from 0.5 to 0.05 as the compression ratio decreases from 30% to 10%. In contrast, 8-shot settings experience a less severe reduction, from 0.75 to 0.5, under the same compression conditions. This suggests that prompts with more shots, by virtue of containing more contextual examples, offer a richer set of reference points for the model. Consequently, the model's reliance on any single example being perfectly preserved in the compressed KV cache is reduced, leading to greater robustness against aggressive compression.

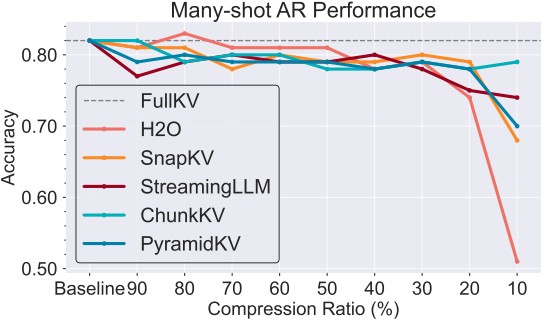

(a) Many-shot *Arithmetic Reasoning* on LLaMA3.1-8B-Instruct

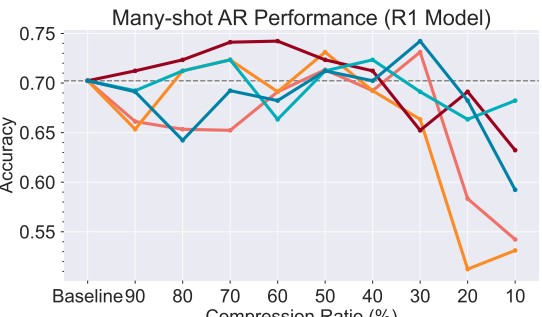

(b) Many-shot *Arithmetic Reasoning* on DeepSeek-R1-Distill-Llama-8B

Figure 6: Many-shot scenario on KV cache compression

*Observation 4.* **Chunk-Level Superiority**: Chunk-level compression is more effective for long-context structured reasoning tasks. Inspired by Agarwal et al. (2024), we consider many-shot in-context learning as a long-context reasoning task, which is more complex than existing long-context benchmarks, such as LongBench and NIAH. Figure 6 shows the performance of KV cache compression methods on a 50-shot GSM8K task, where the prompt length exceeds 4K tokens. From the figure, we observe that ChunkKV Liu et al. (2025) demonstrates the most stability when the compression ratio is below 10% on both LLaMA-3.1-8B-Instruct and DeepSeek-R1-Distill-Llama-8B, indicating that in more complex long-context arithmetic reasoning tasks, chunk-level

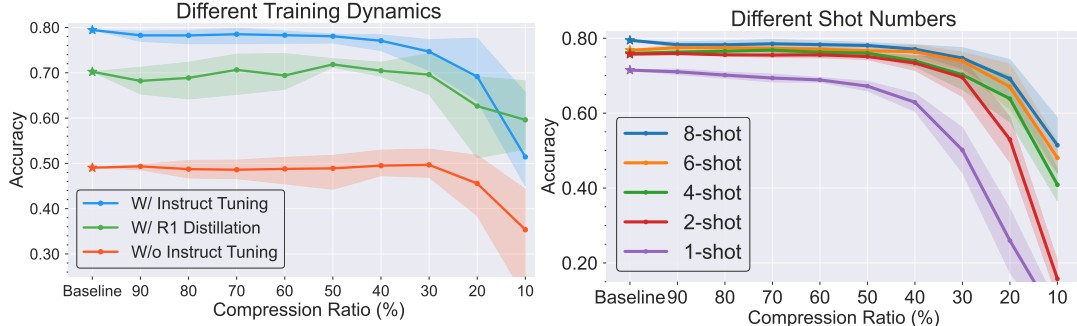

Figure 7: Performance Comparison of KV Cache Compression Methods on different training dynamics on *Arithmetic Reasoning*

Figure 8: Average Performance Across Different Shot Numbers

retention is more effective at preserving semantic information. Highlighting the effectiveness of chunk-level compression for long-context structured reasoning tasks.

*Observation 5.* **Prompt-Gain Sensitivity**: KV cache compression significantly reduces performance gains from ICL and CoT. As shown in Table 2, different tasks exhibit varying levels of performance improvement from zero-shot to CoT prompting. *Arithmetic reasoning* shows a dramatic improvement of 50.41%, while *World Knowledge* demonstrates a more modest gain of 6.20%.

From Figure 4, we find that tasks with larger CoT improvements, such as *Arithmetic reasoning*, are more sensitive to KV cache compression. This suggests that when a task is heavily based on CoT to achieve better performance, compression of these crucial prompt elements has a more substantial impact on model performance. In contrast, tasks like *World Knowledge*,

Table 2: Zero-shot vs Few-shot Performance Comparison

| Benchmark | Zero-shot ↑ | CoT ↑ | Delta∆ |
|---|---|---|---|
| *Arithmetic Reasoning* | 29.04 | 79.45 | +50.41 |
| *World Knowledge* | 62.62 | 68.82 | +6.20 |

where the performance gain from CoT is smaller, show more resilience to KV cache compression, likely because the model relies more on its inherent knowledge than on the specific examples in the prompt.

*Observation 6.* **Long-Context Generation Sensitivity**: KV cache compression exhibits significant performance degradation in long-context generation tasks. As demonstrated in Table 3, our evaluation of three unified compression methods—StreamingLLM, H2O, and PyramidInfer—on *LG-GSM8K* reveals substantial performance limitations. In this arithmetic reasoning task with approximately 4k token generation duration, compression methods show no-

Table 3: KV cache compression methods' performance on *LG-GSM8K*

| Method | 100% | 40% | 35% | 30% | 25% |
|---|---|---|---|---|---|
| FullKV | 46.00 | - | - | - | - |
| StreamingLLM | - | 39.50 | 28.67 | 14.83 | 6.33 |
| H2O | - | 32.66 | 25.17 | 19.83 | 14.83 |
| PyramidInfer | - | 38.33 | 27.67 | 20.50 | 16.67 |
| ShotKV(Ours) | - | **47.33** | **41.33** | **38.33** | **26.83** |

table deterioration, with performance declining by more than 20% at compression ratios below 30%. The ShotKV is our proposed method that aims to improve the performance of KV cache compression on Long-Context Generation tasks, details in Section 4.

# 4 SHOTKV

Our comprehensive empirical investigation in Section 3.3 has systematically revealed critical vulnerabilities in current KV cache compression approaches when applied to a diverse range of fundamental LLM capabilities. Key findings indicate that:

- Specific task categories, notably *Arithmetic Reasoning* (**Observation 1**) and *Long-Context Generation* (**Observation 6**), exhibit pronounced performance degradation under aggressive compression.
- The integrity of prompt information is paramount; tasks that derive significant benefits from ICL and CoT (**Observation 5**) or rely heavily on n-shot prompts (as evidenced by the attention patterns in Figure 2) are particularly susceptible to information loss from compression.

- Preserving semantic coherence is crucial, with chunk-level strategies showing promise in complex reasoning tasks (**Observation 4**), suggesting that compressing or discarding tokens without regard to these semantic units can be detrimental.

These observations collectively underscore the limitations of existing unified compression methods, which often fail to preserve nuanced structured information embedded in prompts, thereby leading to documented performance drops. This necessitates a more discerning compression strategy that is acutely aware of the semantic and structural importance of prompt components, especially for tasks demanding intricate reasoning and extensive generation.

To address these multifaceted challenges identified by our empirical study, we introduce ShotKV, a novel decoding-time compression method. ShotKV is specifically designed to mitigate the observed performance degradation by strategically managing KV cache during the prefill and decoding phases. Our approach is founded on the principle that n-shot examples in prompts are not merely token sequences, but constitute coherent semantic chunks vital for effective reasoning (a concept supported by Figure 2 and Observation 4). We therefore design ShotKV to preserve these shot examples intact during the prefill phase, complemented by a distinct strategy for the decoding phase, aiming for robust performance, particularly on the sensitive tasks highlighted in our analysis.

## 4.1 IMPLEMENTATION

The **ShotKV** (Prefill-Decoding Separated **Shot**-aware **KV** Cache Compression), which separates the compression strategy for prefill and decoding phases. The key insight is that the prefill phase KV cache, which contains crucial prompt information, should be compressed once and remain fixed, while the decoding phase KV cache can be dynamically compressed with different strategies.

Given a prompt with $n$ shots and tokens generated, we define:

$$KV_{\text{total}} = KV_{\text{prefill}} \cup KV_{\text{decoding}} \tag{2}$$

For the prefill phase, we compute shot importance and preserve complete shot examples:

$$\text{Score}_{\text{prefill}}(s_i) = \frac{1}{k_i} \sum_{t \in s_i} \sum_{h=1}^{H} \sum_{l=1}^{L} \alpha_{t,h}^l \tag{3}$$

where $s_i$ represents the $i$-th shot example containing $k_i$ tokens. The term $\alpha_{t,h}^l$ denotes the attention weight assigned by the query vector (corresponding to the first token to be decoded immediately following the prompt) to the key vector of a token $t$ within shot $s_i$, in attention head $h$ at transformer layer $l$. Once the prefill phase KV cache is compressed based on these scores, it remains fixed throughout the generation process.

Given a prefill compression ratio $r_p$, we prioritize shots with higher scores while ensuring that the total number of preserved tokens does not exceed the KV cache limit. Specifically, the shots are ranked by their scores and selected in descending order until they reach the compression budget $r_p \times |KV_{\text{prefill}}|$. This shot-level selection strategy helps to maintain the semantic coherence of important examples while adhering to memory constraints.

$$KV_{\text{prefill}}^C = \text{Compress}(\{s_i | s_i \in S_{\text{preserved}}^*\}) \tag{4}$$

$$\text{where} \quad S_{\text{preserved}} = \underset{S \subseteq \{s_1, \ldots, s_n\}}{\arg\max} \sum_{s_i \in S} \text{Score}_{\text{prefill}}(s_i) \tag{5}$$

$$\text{subject to:} \quad \sum_{s_i \in S} k_i \leq r_p \times |KV_{\text{prefill}}| \tag{6}$$

Here, $KV_{\text{prefill}}^C$ represents the compressed KV cache for prefilling and $S_{\text{preserved}}$ represents the optimal subset of shots that should be preserved after compression. The first equation aims to maximize the total importance score of the selected shots, where $\{s_1, \ldots, s_n\}$ represents all available shots and $\text{Score}_{\text{prefill}}(s_i)$ is the importance score of the shot $s_i$ calculated using attention weights as defined earlier. The second equation enforces the memory constraint: the total number of tokens ($k_i$) in the selected shots must not exceed the allocated budget, which is determined by the prefill compression ratio $r_p$ multiplied by the original KV cache size.

For the decoding phase, we compute importance scores only for the tokens generated during decoding:

$$\text{Score}_{\text{decoding}}(t) = \sum_{h=1}^{H} \sum_{l=1}^{L} \alpha_{t,h}^l \tag{7}$$

Here, for a previously generated token $t$, $\alpha_{t,h}^l$ is similarly defined as the attention weight assigned by the query vector of the current token being generated to the key vector of token $t$, within head $h$ at layer $l$. Thus, $\text{Score}_{\text{decoding}}(t)$ represents the total attention received by token $t$ from the current generation step.

Given a decoding compression ratio $r_d$, we select the tokens with the highest scores to preserve. The compressed decoding KV cache $KV_{\text{decoding}}^C$ retains only the top-$k$ tokens where $k = r_d \times |KV_{\text{decoding}}|$, effectively maintaining the most influential context tokens while reducing memory usage:

$$KV_{\text{decoding}}^C = \text{TopK}(KV_{\text{decoding}}, \text{Score}_{\text{decoding}},$$
$$k = r_d \times |KV_{\text{decoding}}|) \tag{8}$$

Finally, we combine compressed prefill and decoding KV caches to form the total compressed KV cache:

$$KV_{\text{total}} = KV_{\text{prefill}}^C \cup KV_{\text{decoding}}^C \tag{9}$$

### 4.2 EMPIRICAL RESULTS

In this section, we evaluate ShotKV under two scenarios: many-shot *Arithmetic Reasoning* with multiple KV cache compression methods, and *LG-GSM8K* with three unified compression methods that optimize the KV cache during generation. We additionally report a non-ICL generalization study on HotpotQA and an ablation that isolates the contribution of the decoding-phase compression on LG-GSM8K; detailed experimental results are provided in Section C.2 and Section 4.2.

**Baseline.** For *LG-GSM8K* evaluation, we employ three state-of-the-art unified compression methods as baselines: StreamingLLM Xiao et al. (2024), H2O Zhang et al. (2023), and PyramidInfer Yang et al. (2024). We conduct experiments using LLaMA-3-8B-Instruct Dubey et al. (2024) on the *LG-GSM8K* benchmark Liu et al. (2024d), maintaining consistent parameters with Observation 6 ($K = 35$, $T = 20$). For many-shot *Arithmetic Reasoning* experiments, we follow the configuration detailed in Observation 4.

**Main results and analysis.** From the Table 4, we can see that ShotKV achieves the best performance on *LG-GSM8K*, maintaining high performance at low compression ratios. Specifically, at a compression ratio of 40%, ShotKV achieves 47.33% accuracy, surpassing the full kv cache baseline (46.00%) and showing substantial improvements over other methods (32.66%-39.50%). And Table 3 shows that ShotKV also achieves the best performance on many-shot *Arithmetic*

Table 4: KV cache compression methods' performance on Many-shot *Arithmetic Reasoning*

| Method | 100% | 40% | 30% | 20% | 10% |
|---|---|---|---|---|---|
| FullKV | 82.35 | - | - | - | - |
| StreamingLLM | - | 80.37 | 78.35 | 75.37 | 74.32 |
| H2O | - | 78.32 | 79.32 | 74.28 | 51.27 |
| PyramidKV | - | 78.34 | 79.34 | 78.32 | 70.37 |
| SnapKV | - | 79.35 | 80.38 | 79.34 | 68.27 |
| ChunkKV | - | 78.32 | 79.32 | 78.35 | 79.32 |
| ShotKV(Ours) | - | **81.07** | **80.82** | **80.57** | **80.37** |

*Reasoning*, maintaining high performance at low compression ratios. Even in aggressive compression ratios (25%-30%), ShotKV maintains relatively stable performance (26.83%-38.33%), while other methods experience more severe degradation (6.33%-16.67%). This superior performance can be attributed to two key design choices: (1) the preservation of complete shot examples during compression maintains the semantic coherence necessary for mathematical reasoning, and (2) the separation of prefill and decoding phase compression allows for more flexible and task-appropriate token retention strategies. These results suggest that our shot-aware compression strategy is particularly effective for long-context generation tasks that require maintaining complex reasoning chains, such as mathematical problem solving.

**Latency and Throughput**  We further compare the inference efficiency of ShotKV and the FullKV baseline in terms of latency and throughput under different input and output sequence lengths. As shown in Table 5, ShotKV consistently reduces latency and improves throughput compared to FullKV. For example, with an input length of 8192 and output length of 4096, ShotKV achieves an 11.3% reduction in latency and a 13.1% increase in throughput. These results demonstrate that ShotKV not only maintains model performance under aggressive KV cache compression, but also brings tangible efficiency benefits for long-context inference.

Table 5: Latency and throughput comparison between ShotKV and FullKV under different input-output configurations. Percentages in parentheses indicate improvements over FullKV baseline. The experiments test on the A40 server with batch size 1.

| Method | Sequence Length | | Performance Metrics | |
|---|---|---|---|---|
| | Input | Output | Latency(s) ↓ | Throughput(T/S) ↑ |
| FullKV | 4096 | 4096 | 175.50 | 37.73 |
| ShotKV | 4096 | 4096 | 162.85 (7.2%) | 41.12 (9.0%) |
| FullKV | 8192 | 4096 | 183.42 | 55.93 |
| ShotKV | 8192 | 4096 | 162.78 (11.3%) | 63.24 (13.1%) |

**Generalization to Non-ICL Tasks (HotpotQA).**  For a document QA setting without few-shot ICL, we adapt ShotKV by treating each sentence as a coherent semantic unit (analogous to a *shot*). Even under an aggressive 10% compression ratio on LLaMA-3-8B-Instruct, ShotKV remains competitive with the best-performing method, as shown in Table 6

Table 6: LLaMA-3-8B-Instruct on HotpotQA at 10% compression.

| Method | Score |
|---|---|
| FullKV | 45.55 |
| StreamingLLM | 40.27 |
| H2O | 40.84 |
| SnapKV | 43.36 |
| PyramidKV | 43.80 |
| ChunkKV | 43.27 |
| ShotKV (Ours) | 43.60 |

## 5 CONCLUSION

This paper presents KVFundaBench, a benchmark for systematically evaluating the effects of KV cache compression on various fundamental LLM capabilities. Our findings reveal that performance degradation is highly task dependent, with arithmetic reasoning and long-context generation being particularly sensitive (*Task-Dependent Degradation* and *Long-Context Generation Sensitivity*). We also highlight that compression sensitivity is influenced by a confluence of factors, including inherent model characteristics such as training dynamics (*Model-Type Robustness*), prompt-level attributes like length (*Prompt Length Vulnerability*), and the reliance on in-context examples (*Prompt-Gain Sensitivity*). Crucially, we demonstrate the importance of preserving the semantic integrity of prompt components, especially at a chunk or shot level, for complex reasoning and generation tasks where current methods often struggle and where chunk-based approaches show promise (*Chunk-Level Superiority*).

Based on these insights, we introduced ShotKV, a novel compression framework that distinctively manages prefill and decoding phases while prioritizing shot-level semantic coherence to mitigate information loss in sensitive tasks. ShotKV demonstrates superior performance, notably on long-context arithmetic reasoning and generation tasks, maintaining high accuracy even at aggressive compression ratios. The results of KVFundaBench and the efficacy of ShotKV underscore the potential for more nuanced compression strategies and suggest promising future research avenues.

## ETHICS STATEMENT

This work focuses on the technical advancement of LLM efficiency. Our goal is to reduce the computational and energy costs of LLMs, thereby making AI technology more accessible and sustainable. We built our benchmark using public academic datasets and foresee no direct negative societal

impacts. While we acknowledge the broader societal implications of advancing AI capabilities, our work is intended to contribute positively to the research community by enabling more efficient model deployment.

## REPRODUCIBILITY STATEMENT

To ensure our results are reproducible, we will release all code for our method, ShotKV, and evaluation scripts. Our experiments exclusively use publicly available models (e.g., LLaMA-3.1, Mistral-7B) and standard academic datasets (e.g., MMLU, GSM8K), all evaluated using the open-source `lm-evaluation-harness` and `KVpress` framework. Detailed hyperparameters and specific experimental configurations are provided in Appendix **??**.

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

APPENDIX

## A   USE OF LLMs IN PAPER WRITING

We used LLMs solely to aid and polish the writing (e.g., wording refinement and grammar), without generating or altering experimental designs, methods, results, or conclusions. All technical content, analyses, figures, and tables were authored and verified by the researchers.

## B   RELATED WORK

**Key–value Cache Optimization Techniques**   KV cache is the core component in LLM inference, which avoids repetitive computations by caching Key and Value vectors. However, the cost of caching KV increases exponentially with the expansion of the model size and the length of the context Pope et al. (2023). Some approaches have been published to alleviate the problem. For example, KV Compression designs efficient content selection strategies to filter and manage tokens Zhang et al. (2023); Adnan et al. (2024). Some methods identify important tokens by focusing on high attention allocation Li et al. (2024b), while others optimize token selection by combining attention scores with value vector norms to improve importance evaluation Guo et al. (2024). Techniques like Pyramid-Infer reduce critical contexts layer by layer based on the distribution of attention scores Yang et al. (2024), and StreamingLLM preserves attention sinks to maintain stable performance in extended sequences Xiao et al. (2024). Researchers reduce storage costs by merging similar context representations and solving input disturbances caused by compression Zhang et al. (2024b). For example, CaM Zhang et al. (2024b) works by integrating the KV cache to be dropped into the retained cache in proportion to the attention weight. In addition, Yao et al. (2024) proposes CacheBlend to achieve a selective KV recompute. Only partial KVs of crucial tokens are updated to reduce the delay in the prefill stage and increase the throughput. In addition, the dynamic budget allocation method is also used to optimize the KV cache, which adjusts the resource allocation in real time according to the importance of the context, providing a balance between performance and efficiency in multitask inference scenarios Cai et al. (2024); Feng et al. (2024); Kim et al. (2025).Wu et al. (2024) proposes a prefill-decoding separation strategy to optimize the KV cache compression.

**Evaluation of LLMs' Fundamental Abilities**   Accurately evaluating the fundamental capabilities of large language models is crucial to understand their true potential and limitations. The evaluation typically spans across several key dimensions: world knowledge tasks like MMLU Hendrycks et al. (2020),BBH Suzgun et al. (2022) assess models' grasp of diverse domains through multiple-choice questions; commonsense reasoning tasks such as CSQA Talmor et al. (2019) evaluate inference and context understanding abilities; arithmetic reasoning benchmarks like GSM8K Cobbe et al. (2021) test mathematical problem-solving capabilities through step-by-step reasoning; code generation tasks including HumanEval Chen et al. (2021) measure the ability to generate functionally correct code; and safety evaluations using benchmarks like JailBreakV Luo et al. (2024) assess models' robustness against harmful content generation. Additionally, long-context benchmarks such as Long-Bench Bai et al. (2023; 2025) and Need-In-A-Haystack (NIAH) Kamradt (2023) aiming to evaluate models' long-context summarization and retrieval capabilities. Furthermore, LongGenBench Liu et al. (2024d) evaluates the models' ability to process and generate responses for extended input sequences. And recently, in-context many-shot learning has been recognized as a long-context reasoning paradigm Agarwal et al. (2024), which considers the number of shots as a critical factor in the performance of LLM. Although these tasks typically employ automatic evaluation metrics for standardization, KV cache compression may introduce unique challenges, particularly in tasks requiring complex reasoning chains or extensive knowledge retrieval.

**KV cache sharing**   Recent work has explored various strategies for sharing KV caches across transformer layers. The Layer Condensed KV Cache (LCKV) (Wu & Tu, 2024) computes the KV only for the top layer and pairs them with queries from all layers, while optionally retaining standard attention for a few top and bottom layers to mitigate performance degradation. Similarly, You Only Cache Once (YOCO) (Sun et al., 2024) computes KVs exclusively for the top layer but pairs them with queries from only the top half of layers, employing efficient attention in the bottom layers to maintain a constant cache size. In contrast, Cross-Layer Attention (CLA) (Brandon et al., 2024) divides layers into groups, pairing queries from all layers in each group with KVs from that group's bottom layer. MiniCache (Liu et al., 2024b) introduces a novel method that merges KV caches in layering while enabling recovery during compute-in-place operations, optimizing the size of the KV

cache. These methods illustrate various trade-offs between computation, memory usage, and model performance when sharing KV caches across transformer layers.

**Prompting Compression**   Recent advances in prompt compression have yielded innovative approaches to information density optimization in natural language processing. Research by Wingate et al. (2022) demonstrates how soft prompting techniques can achieve higher information density per token. Building upon this foundation, AutoCompressor (Chevalier et al., 2023) leverages soft prompts to both condense input sequences and expand model context windows. Parallel developments by Zhou et al. (2023) and Wang et al. (2023) showcase iterative summarization strategies using LLMs, establishing persistent memory mechanisms particularly beneficial for narrative construction and conversational systems. The progressive development of the LLMLingua framework (Jiang et al., 2023b; 2024; Fei et al., 2024) has advanced prompt compression capabilities across extended context processing, logical reasoning, and retrieval-augmented generation. Notable contributions from Fei et al. (2024) demonstrate effective context management through automated segmentation and semantic condensation using pre-trained language models.

**General Tasks**   General tasks refer to evaluating the overall performance of LLMs under mathematical inference, logic reasoning, and common knowledge. GSM8K Cobbe et al. (2021) and MMLU Hendrycks et al. (2020) are representative tasks. The former focuses on the step-by-step reasoning ability of mathematical problem solving, while the latter covers assessment of common sense and expertise in multiple areas. Besides, MATH Hendrycks et al. (2021) spans various mathematical fields, ranging from elementary algebra to calculus, aiming to improve the mathematical problem-solving capabilities of LLMs. Meanwhile, MathQA Amini et al. (2019) is a large-scale dataset comprising approximately 37,000 multiple-choice questions with precise annotations, designed to enhance the interpretability and performance of LLMs. In addition, BBH Suzgun et al. (2022), a subset of BIG-Bench Srivastava et al. (2022), focuses on challenging tasks. BBH includes multi-step reasoning problems, highlighting the importance of Chain-of-Thought prompting in LLMs. Similarly, CSQA Talmor et al. (2019) is a task that combines knowledge graph-based multi-step reasoning with conversational capabilities. CSQA emphasizes inference and context understanding grounded in knowledge graphs. Normally, the general tasks apply automatic evaluation metrics (e.g. multi-choice accuracy) to ensure comparability and standardization. However, optimization strategies like KV cache compression may introduce challenges in executing the mentioned tasks. Filtering and dropping of contexts are involved in the compression strategy which may lead to an intermediate inference steps missing. In addition, in tasks such as MMLU that are highly dependent on knowledge coverage, compression may weaken the model's ability to capture long context or rare domain knowledge Yuan et al. (2024).

**Security Tasks**   Security tasks focus on assessing the robustness and protections of LLMs against harmful content, including truthfulness Lin et al. (2021), toxicity Hartvigsen et al. (2022), and bias Liang et al. (2021). Recently, researchers noticed the weakness of LLMs in adversarial prompts Zhu et al. (2023), especially in generating illegal or inappropriate content under jailbreak prompts. Shen et al. (2024) analyze the jailbreak prompts in real cases to reveal the failure of model security mechanism under complex malicious input. Meanwhile, Deng et al. (2023) demonstrates the multilingual jailbreak makes model security in low-resource languages easier to bypass, significantly increasing the probability that users of low-resource languages will generate insecure content. Similar to general tasks, KV optimization techniques can cause the model to ignore potential security threats when dealing with jailbreak prompts, thereby improving the success rate of adversarial prompts Li et al. (2024a).

**Code Generation Tasks**   Code generation tasks test the capacities of LLMs to generate code, which not only requires that the model can generate syntactic code based on natural language description but also has certain logical reasoning abilities. HumanEval Chen et al. (2021) and MBPP Austin et al. (2021) are the commonly used benchmarks. They measure the functional correctness of the model by testing the results of the code's execution.

**Long-context Tasks**   Recent developments in evaluating long-context models have produced a comprehensive ecosystem of benchmarks, focusing on both comprehension depth and retrieval efficiency. In the comprehension domain, $\infty$-Bench (Zhang et al., 2024a) has established new

standards by crafting evaluation scenarios exceeding 100,000 tokens, while LongBench (Bai et al., 2023; 2025) introduced multilingual assessment frameworks spanning document comprehension, text synthesis, and programming tasks. Further enriching this landscape, ZeroSCROLLS (Shaham et al., 2023) and L-Eval (An et al., 2023) have expanded evaluation criteria to encompass real-world applications, particularly in query-based content summarization. The emergence of many-shot learning as a distinct paradigm for extended context processing Agarwal et al. (2024) has added another dimension to this field. Notable contributions from LongGenBench Liu et al. (2024d) have advanced evaluation methodologies by combining extensive response generation requirements with efficient, cost-effective quality metrics.

The development of retrieval-focused benchmarks has taken a distinct approach, predominantly utilizing constructed datasets that enable precise experimental control, particularly in managing input sequence lengths. This methodology helps neutralize variations in model performance stemming from differences in training approaches. Substantial research efforts have yielded specialized synthetic frameworks for assessing retrieval capabilities (Kamradt, 2023; Mohtashami & Jaggi, 2023; Li et al., 2023; Liu et al., 2024c; Hsieh et al., 2024), while concurrent investigations have revealed the broader implications of extended context processing for enhanced reasoning capabilities (Tay et al., 2021).

## C EXPERIMENT DETAILS

### C.1 DETAIL RESULTS

This section provide the detailed results of experiments in this paper, the results are shown in the format of $x_y$, where $x$ is the performance of the method and $y$ is the $\Delta P$ from the Equation (1).

**Observation 1. KV cache compression methods show task-dependent performance degradation, WK and CSR are more robust to KV cache compression.**

The detailed results of different KV cache compression methods are shown in Table 8, different tasks exhibit notably varied sensitivities to KV cache compression, particularly under aggressive compression ratios. At a 10% compression ratio, MMLU demonstrates remarkable resilience with less than 1% average performance degradation, while GSM8K experiences a severe average performance drop exceeding 35%. Other tasks show moderate to significant degradation, ranging from 6.5% to 17.2%. This substantial variation in compression sensitivity across tasks suggests that the effectiveness of KV cache compression is highly task-dependent, necessitating careful consideration of the specific task requirements when determining appropriate compression ratios.

The Table 7 compares the performance of R1-Distill-Llama-8B and LLaMA-3.1-8B-Instruct under different compression ratios. R1-Distill-Llama-8B demonstrates more robust performance under compression compared to LLaMA-3.1-8B-Instruct. While both models start with similar baseline performance (0.6938 vs 0.7945), R1-Distill shows significantly less performance degradation under aggressive compression. Specifically, at 30% compression ratio, R1-Distill maintains a performance of 0.6407 (-7.66%), while LLaMA-3.1-8B-Instruct drops to 0.7469 (-6.00%). The difference becomes more pronounced at 10% compression ratio, where R1-Distill achieves 0.5840 (-15.82%) compared to LLaMA-3.1-8B-Instruct's sharp decline to 0.5143 (-35.30%). This suggests that the multi-step reasoning capabilities of R1-Distill contribute to its resilience against aggressive KV cache compression, particularly in maintaining reasoning coherence under limited context conditions.

On safety-focused evaluations, we observe that aggressive compression can disproportionately degrade performance, plausibly because compression may discard or fragment subtle safety-critical keywords and phrases present in system prompts; this disruption can weaken safety constraints during generation.

**Observation 2. Multi-step reasoning LLMs are more robust to KV cache compression.** As shown in Table 9, while instruct-tuned models achieve superior baseline performance (0.7945 vs 0.5122), they demonstrate heightened sensitivity to KV cache compression. This sensitivity becomes particularly pronounced at aggressive compression ratios. At 10% compression ratio, instruct-tuned models suffer an average performance degradation of 35.3% (from 0.7945 to 0.5143), nearly double the degradation observed in non-instruct-tuned models which show a 17.2% drop (from 0.5122 to 0.4244). In contrast, R1-Distill-Llama-8B shows better resilience to compression, with only a

Table 7: Performance Comparison of Different KV Cache Compression Methods on Instruction-Tuning Model and Multi-Step Reasoning Model

| Benchmark | Ratio | StreamingLLM | H2O | SnapKV | PyramidKV | ChunkKV | Average ↑ |
|---|---|---|---|---|---|---|---|
| | Baseline | R1-Distill-Llama-8B FullKV: 0.6938 | | | | | |
| R1-AR | 90% | $0.7167_{(+3.30\%)}$ | $0.6900_{(-0.55\%)}$ | $0.6933_{(-0.07\%)}$ | $0.7100_{(+2.34\%)}$ | $0.6867_{(-1.02\%)}$ | $0.6993_{(+0.79\%)}$ |
| | 80% | $0.6867_{(-1.02\%)}$ | $0.6933_{(-0.07\%)}$ | $0.6933_{(-0.07\%)}$ | $0.7067_{(+1.86\%)}$ | $0.6767_{(-2.47\%)}$ | $0.6913_{(-0.36\%)}$ |
| | 70% | $0.6933_{(-0.07\%)}$ | $0.6633_{(-4.40\%)}$ | $0.7100_{(+2.34\%)}$ | $0.7100_{(+2.34\%)}$ | $0.7000_{(+0.89\%)}$ | $0.6953_{(+0.22\%)}$ |
| | 60% | $0.6833_{(-1.51\%)}$ | $0.6900_{(-0.55\%)}$ | $0.6900_{(-0.55\%)}$ | $0.7133_{(+2.81\%)}$ | $0.7067_{(+1.86\%)}$ | $0.6967_{(+0.42\%)}$ |
| | 50% | $0.6700_{(-3.43\%)}$ | $0.6967_{(+0.42\%)}$ | $0.7067_{(+1.86\%)}$ | $0.7000_{(+0.89\%)}$ | $0.6867_{(-1.02\%)}$ | $0.6920_{(-0.26\%)}$ |
| | 40% | $0.6767_{(-2.47\%)}$ | $0.6800_{(-1.99\%)}$ | $0.5967_{(-13.99\%)}$ | $0.6967_{(+0.42\%)}$ | $0.7133_{(+2.81\%)}$ | $0.6727_{(-3.04\%)}$ |
| | 30% | $0.6600_{(-4.87\%)}$ | $0.5900_{(-14.96\%)}$ | $0.5833_{(-15.93\%)}$ | $0.6700_{(-3.43\%)}$ | $0.7000_{(+0.89\%)}$ | $0.6407_{(-7.66\%)}$ |
| | 20% | $0.6200_{(-10.64\%)}$ | $0.4933_{(-28.90\%)}$ | $0.5633_{(-18.81\%)}$ | $0.6833_{(-1.51\%)}$ | $0.6533_{(-5.84\%)}$ | $0.6026_{(-13.14\%)}$ |
| | 10% | $0.5167_{(-25.53\%)}$ | $0.5567_{(-19.76\%)}$ | $0.5767_{(-16.88\%)}$ | $0.6267_{(-9.67\%)}$ | $0.6433_{(-7.28\%)}$ | $0.5840_{(-15.82\%)}$ |
| | Baseline | LLaMA-3.1-8B-Instruct FullKV: 0.7945 | | | | | |
| AR | 90% | $0.7695_{(-3.10\%)}$ | $0.7923_{(-0.30\%)}$ | $0.7839_{(-1.30\%)}$ | $0.7854_{(-1.10\%)}$ | $0.7824_{(-1.50\%)}$ | $0.7827_{(-1.50\%)}$ |
| | 80% | $0.7642_{(-3.80\%)}$ | $0.7938_{(-0.10\%)}$ | $0.7824_{(-1.50\%)}$ | $0.7900_{(-0.60\%)}$ | $0.7824_{(-1.50\%)}$ | $0.7826_{(-1.50\%)}$ |
| | 70% | $0.7642_{(-3.80\%)}$ | $0.7900_{(-0.60\%)}$ | $0.7923_{(-0.30\%)}$ | $0.7983_{(+0.50\%)}$ | $0.7809_{(-1.70\%)}$ | $0.7851_{(-1.20\%)}$ |
| | 60% | $0.7650_{(-3.70\%)}$ | $0.7809_{(-1.70\%)}$ | $0.7885_{(-0.80\%)}$ | $0.7923_{(-0.30\%)}$ | $0.7885_{(-0.80\%)}$ | $0.7830_{(-1.50\%)}$ |
| | 50% | $0.7657_{(-3.60\%)}$ | $0.7854_{(-1.10\%)}$ | $0.7847_{(-1.20\%)}$ | $0.7854_{(-1.10\%)}$ | $0.7824_{(-1.50\%)}$ | $0.7807_{(-1.70\%)}$ |
| | 40% | $0.7491_{(-5.70\%)}$ | $0.7688_{(-3.20\%)}$ | $0.7756_{(-2.40\%)}$ | $0.7839_{(-1.30\%)}$ | $0.7763_{(-2.30\%)}$ | $0.7707_{(-3.00\%)}$ |
| | 30% | $0.7051_{(-11.20\%)}$ | $0.7225_{(-9.10\%)}$ | $0.7619_{(-4.10\%)}$ | $0.7718_{(-2.90\%)}$ | $0.7733_{(-2.70\%)}$ | $0.7469_{(-6.00\%)}$ |
| | 20% | $0.6384_{(-19.70\%)}$ | $0.6406_{(-19.40\%)}$ | $0.6884_{(-13.40\%)}$ | $0.7142_{(-10.10\%)}$ | $0.7763_{(-2.30\%)}$ | $0.6916_{(-13.00\%)}$ |
| | 10% | $0.4784_{(-39.80\%)}$ | $0.4503_{(-43.30\%)}$ | $0.5034_{(-36.60\%)}$ | $0.4829_{(-39.20\%)}$ | $0.6566_{(-17.40\%)}$ | $0.5143_{(-35.30\%)}$ |

15.82% performance drop (from 0.6938 to 0.5840) at 10% compression ratio. This pattern suggests that while instruction tuning enhances model capabilities, it also makes the model more dependent on maintaining complete context information. However, models trained with multi-step reasoning capabilities like R1-Distill demonstrate better robustness against aggressive compression, likely due to their enhanced ability to maintain reasoning coherence even with limited context. We hypothesize that the reinforcement learning objective that explicitly incentivizes multi-step reasoning in DeepSeek-R1 yields more structured and robust internal representations of reasoning chains, making them less fragile to KV cache compression.

**Observation 3.  Short prompt length is more sensitive to KV cache compression.** As demonstrated in Table 10, the impact of KV cache compression varies significantly with the number of shots in the prompt. One-shot prompts show extreme vulnerability to aggressive compression, with performance plummeting from 0.7149 to 0.0452 (a 93.7% drop) at 10% compression ratio. This sensitivity gradually decreases as the number of shots increases. For instance, at the same compression ratio, 4-shot prompts show a 46.2% performance drop (from 0.7597 to 0.4088), while 8-shot prompts demonstrate relatively better resilience with a 35.3% reduction (from 0.7945 to 0.5143). This pattern suggests that longer prompts with more examples provide redundancy that helps maintain model performance under compression, while shorter prompts lack this buffer against information loss.

**Observation 4.  Chunk-level compression is more effective for long-context structured reasoning tasks.** As shown in Table 11, ChunkKV demonstrates superior robustness across different compression ratios, particularly under aggressive compression settings. While other methods show significant performance degradation at 10% compression ratio (StreamingLLM: -9.8%, H2O: -37.8%, SnapKV: -17.1%, PyramidKV: -14.6%), ChunkKV maintains relatively stable performance with only a -3.7% drop. This stark contrast in performance suggests that chunk-level compression better preserves the essential contextual information needed for complex reasoning tasks. The method's effectiveness likely stems from its ability to maintain the structural integrity of related context segments, which is particularly crucial for tasks requiring extended logical reasoning and arithmetic operations.

## C.2 Ablation: Prefill-only vs. Full ShotKV on LG-GSM8K.

To assess the contribution of the decoding-phase compression, we ablate it by retaining only prefill compression. As summarized in Table 12, this prefill-only variant substantially underperforms the full method across compression ratios, confirming the importance of the prefill–decoding separation.

Table 8: Performance Comparison of Different KV Cache Compression Methods on KVFundaBench

| Benchmark | Ratio | StreamingLLM | H2O | SnapKV | PyramidKV | ChunkKV | Average ↑ |
|---|---|---|---|---|---|---|---|
| | Baseline | | | FullKV: 0.6882 | | | |
| WK | 90% | $0.6882_{(+0.00\%)}$ | $0.6882_{(+0.00\%)}$ | $0.6882_{(+0.00\%)}$ | $0.6882_{(+0.00\%)}$ | $0.6882_{(+0.00\%)}$ | $0.6882_{(+0.00\%)}$ |
| | 80% | $0.6882_{(+0.00\%)}$ | $0.6882_{(+0.00\%)}$ | $0.6882_{(+0.00\%)}$ | $0.6882_{(+0.00\%)}$ | $0.6882_{(+0.00\%)}$ | $0.6882_{(+0.00\%)}$ |
| | 70% | $0.6881_{(-0.01\%)}$ | $0.6882_{(+0.00\%)}$ | $0.6882_{(+0.00\%)}$ | $0.6882_{(+0.00\%)}$ | $0.6882_{(+0.00\%)}$ | $0.6882_{(+0.00\%)}$ |
| | 60% | $0.6881_{(-0.01\%)}$ | $0.6882_{(+0.00\%)}$ | $0.6882_{(+0.00\%)}$ | $0.6882_{(+0.00\%)}$ | $0.6882_{(+0.00\%)}$ | $0.6882_{(+0.00\%)}$ |
| | 50% | $0.6881_{(-0.01\%)}$ | $0.6882_{(+0.00\%)}$ | $0.6882_{(+0.00\%)}$ | $0.6882_{(+0.00\%)}$ | $0.6882_{(+0.00\%)}$ | $0.6882_{(+0.00\%)}$ |
| | 40% | $0.6879_{(-0.04\%)}$ | $0.6882_{(+0.00\%)}$ | $0.6882_{(+0.00\%)}$ | $0.6882_{(+0.00\%)}$ | $0.6882_{(+0.00\%)}$ | $0.6881_{(-0.01\%)}$ |
| | 30% | $0.6876_{(-0.09\%)}$ | $0.6880_{(-0.03\%)}$ | $0.6880_{(-0.03\%)}$ | $0.6882_{(+0.00\%)}$ | $0.6882_{(+0.00\%)}$ | $0.6880_{(-0.03\%)}$ |
| | 20% | $0.6859_{(-0.33\%)}$ | $0.6878_{(-0.06\%)}$ | $0.6880_{(-0.03\%)}$ | $0.6882_{(+0.00\%)}$ | $0.6882_{(+0.00\%)}$ | $0.6876_{(-0.08\%)}$ |
| | 10% | $0.6787_{(-1.38\%)}$ | $0.6852_{(-0.44\%)}$ | $0.6831_{(-0.74\%)}$ | $0.6882_{(0.00\%)}$ | $0.6842_{(-0.58\%)}$ | $0.6839_{(-0.63\%)}$ |
| | Baseline | | | FullKV: 0.7945 | | | |
| AR | 90% | $0.7695_{(-3.10\%)}$ | $0.7923_{(-0.30\%)}$ | $0.7839_{(-1.30\%)}$ | $0.7854_{(-1.10\%)}$ | $0.7824_{(-1.50\%)}$ | $0.7827_{(-1.50\%)}$ |
| | 80% | $0.7642_{(-3.80\%)}$ | $0.7938_{(-0.10\%)}$ | $0.7824_{(-1.50\%)}$ | $0.7900_{(-0.60\%)}$ | $0.7824_{(-1.50\%)}$ | $0.7826_{(-1.50\%)}$ |
| | 70% | $0.7642_{(-3.80\%)}$ | $0.7900_{(-0.60\%)}$ | $0.7923_{(-0.30\%)}$ | $0.7983_{(+0.50\%)}$ | $0.7809_{(-1.70\%)}$ | $0.7851_{(-1.20\%)}$ |
| | 60% | $0.7650_{(-3.70\%)}$ | $0.7809_{(-1.70\%)}$ | $0.7885_{(-0.80\%)}$ | $0.7923_{(-0.30\%)}$ | $0.7885_{(-0.80\%)}$ | $0.7830_{(-1.50\%)}$ |
| | 50% | $0.7657_{(-3.60\%)}$ | $0.7854_{(-1.10\%)}$ | $0.7847_{(-1.20\%)}$ | $0.7854_{(-1.10\%)}$ | $0.7824_{(-1.50\%)}$ | $0.7807_{(-1.70\%)}$ |
| | 40% | $0.7491_{(-5.70\%)}$ | $0.7688_{(-3.20\%)}$ | $0.7756_{(-2.40\%)}$ | $0.7839_{(-1.30\%)}$ | $0.7763_{(-2.30\%)}$ | $0.7707_{(-3.00\%)}$ |
| | 30% | $0.7051_{(-11.20\%)}$ | $0.7225_{(-9.10\%)}$ | $0.7619_{(-4.10\%)}$ | $0.7718_{(-2.90\%)}$ | $0.7733_{(-2.70\%)}$ | $0.7469_{(-6.00\%)}$ |
| | 20% | $0.6384_{(-19.70\%)}$ | $0.6406_{(-19.40\%)}$ | $0.6884_{(-13.40\%)}$ | $0.7142_{(-10.10\%)}$ | $0.7763_{(-2.30\%)}$ | $0.6916_{(-13.00\%)}$ |
| | 10% | $0.4784_{(-39.80\%)}$ | $0.4503_{(-43.30\%)}$ | $0.5034_{(-36.60\%)}$ | $0.4829_{(-39.20\%)}$ | $0.6566_{(-17.40\%)}$ | $0.5143_{(-35.30\%)}$ |
| | Baseline | | | FullKV: 0.7748 | | | |
| CSR | 90% | $0.7748_{(+0.00\%)}$ | $0.7748_{(+0.00\%)}$ | $0.7748_{(+0.00\%)}$ | $0.7748_{(+0.00\%)}$ | $0.7748_{(+0.00\%)}$ | $0.7748_{(+0.00\%)}$ |
| | 80% | $0.7748_{(+0.00\%)}$ | $0.7748_{(+0.00\%)}$ | $0.7748_{(+0.00\%)}$ | $0.7748_{(+0.00\%)}$ | $0.7748_{(+0.00\%)}$ | $0.7748_{(+0.00\%)}$ |
| | 70% | $0.7748_{(+0.00\%)}$ | $0.7748_{(+0.00\%)}$ | $0.7748_{(+0.00\%)}$ | $0.7748_{(+0.00\%)}$ | $0.7748_{(+0.00\%)}$ | $0.7748_{(+0.00\%)}$ |
| | 60% | $0.7748_{(+0.00\%)}$ | $0.7748_{(+0.00\%)}$ | $0.7748_{(+0.00\%)}$ | $0.7748_{(+0.00\%)}$ | $0.7748_{(+0.00\%)}$ | $0.7748_{(+0.00\%)}$ |
| | 50% | $0.7748_{(+0.00\%)}$ | $0.7748_{(+0.00\%)}$ | $0.7748_{(+0.00\%)}$ | $0.7748_{(+0.00\%)}$ | $0.7748_{(+0.00\%)}$ | $0.7748_{(+0.00\%)}$ |
| | 40% | $0.7748_{(+0.00\%)}$ | $0.7748_{(+0.00\%)}$ | $0.7748_{(+0.00\%)}$ | $0.7748_{(+0.00\%)}$ | $0.7748_{(+0.00\%)}$ | $0.7748_{(+0.00\%)}$ |
| | 30% | $0.7748_{(+0.00\%)}$ | $0.7748_{(+0.00\%)}$ | $0.7748_{(+0.00\%)}$ | $0.7748_{(+0.00\%)}$ | $0.7748_{(+0.00\%)}$ | $0.7748_{(+0.00\%)}$ |
| | 20% | $0.7174_{(-7.40\%)}$ | $0.7748_{(+0.00\%)}$ | $0.7740_{(-0.10\%)}$ | $0.7748_{(+0.00\%)}$ | $0.7699_{(-0.60\%)}$ | $0.7622_{(-1.60\%)}$ |
| | 10% | $0.6806_{(-12.20\%)}$ | $0.7510_{(-3.10\%)}$ | $0.7191_{(-7.20\%)}$ | $0.7723_{(-0.30\%)}$ | $0.7002_{(-9.60\%)}$ | $0.7246_{(-6.50\%)}$ |
| | Baseline | | | FullKV: 0.8895 | | | |
| SA | 90% | $0.8893_{(-0.00\%)}$ | $0.8890_{(-0.10\%)}$ | $0.8894_{(-0.00\%)}$ | $0.8893_{(-0.00\%)}$ | $0.8896_{(+0.00\%)}$ | $0.8893_{(-0.00\%)}$ |
| | 80% | $0.8878_{(-0.20\%)}$ | $0.8885_{(-0.10\%)}$ | $0.8895_{(+0.00\%)}$ | $0.8891_{(-0.00\%)}$ | $0.8894_{(-0.00\%)}$ | $0.8889_{(-0.10\%)}$ |
| | 70% | $0.8872_{(-0.30\%)}$ | $0.8879_{(-0.20\%)}$ | $0.8896_{(+0.00\%)}$ | $0.8889_{(-0.10\%)}$ | $0.8895_{(+0.00\%)}$ | $0.8886_{(-0.10\%)}$ |
| | 60% | $0.8845_{(-0.60\%)}$ | $0.8848_{(-0.50\%)}$ | $0.8892_{(-0.00\%)}$ | $0.8887_{(-0.10\%)}$ | $0.8899_{(+0.00\%)}$ | $0.8874_{(-0.20\%)}$ |
| | 50% | $0.8849_{(-0.50\%)}$ | $0.8749_{(-1.60\%)}$ | $0.8886_{(-0.10\%)}$ | $0.8884_{(-0.10\%)}$ | $0.8894_{(-0.00\%)}$ | $0.8852_{(-0.50\%)}$ |
| | 40% | $0.8734_{(-1.80\%)}$ | $0.8557_{(-3.80\%)}$ | $0.8880_{(-0.20\%)}$ | $0.8877_{(-0.20\%)}$ | $0.8900_{(+0.10\%)}$ | $0.8790_{(-1.20\%)}$ |
| | 30% | $0.8329_{(-6.40\%)}$ | $0.8015_{(-9.90\%)}$ | $0.8858_{(-0.40\%)}$ | $0.8899_{(+0.00\%)}$ | $0.8846_{(-0.60\%)}$ | $0.8589_{(-3.50\%)}$ |
| | 20% | $0.6501_{(-26.90\%)}$ | $0.7178_{(-19.30\%)}$ | $0.8806_{(-1.00\%)}$ | $0.8751_{(-1.60\%)}$ | $0.8902_{(+0.10\%)}$ | $0.8028_{(-9.70\%)}$ |
| | 10% | $0.5314_{(-40.30\%)}$ | $0.6544_{(-26.40\%)}$ | $0.8434_{(-5.20\%)}$ | $0.8556_{(-3.80\%)}$ | $0.8799_{(-1.10\%)}$ | $0.7529_{(-15.40\%)}$ |
| | Baseline | | | FullKV: 0.5122 | | | |
| CG | 90% | $0.5061_{(-1.20\%)}$ | $0.5183_{(+1.20\%)}$ | $0.5122_{(+0.00\%)}$ | $0.5122_{(+0.00\%)}$ | $0.5122_{(+0.00\%)}$ | $0.5122_{(+0.00\%)}$ |
| | 80% | $0.5061_{(-1.20\%)}$ | $0.5183_{(+1.20\%)}$ | $0.5183_{(+1.20\%)}$ | $0.5305_{(+3.60\%)}$ | $0.5061_{(-1.20\%)}$ | $0.5159_{(+0.70\%)}$ |
| | 70% | $0.5000_{(-2.40\%)}$ | $0.5244_{(+2.40\%)}$ | $0.5122_{(+0.00\%)}$ | $0.5183_{(+1.20\%)}$ | $0.5122_{(+0.00\%)}$ | $0.5134_{(+0.20\%)}$ |
| | 60% | $0.5061_{(-1.20\%)}$ | $0.5366_{(+4.80\%)}$ | $0.5366_{(+4.80\%)}$ | $0.5305_{(+3.60\%)}$ | $0.5244_{(+2.40\%)}$ | $0.5268_{(+2.90\%)}$ |
| | 50% | $0.4939_{(-3.60\%)}$ | $0.5427_{(+6.00\%)}$ | $0.5061_{(-1.20\%)}$ | $0.4939_{(-3.60\%)}$ | $0.4878_{(-4.80\%)}$ | $0.5049_{(-1.40\%)}$ |
| | 40% | $0.4817_{(-6.00\%)}$ | $0.5427_{(+6.00\%)}$ | $0.5244_{(+2.40\%)}$ | $0.4939_{(-3.60\%)}$ | $0.5000_{(-2.40\%)}$ | $0.5085_{(-0.70\%)}$ |
| | 30% | $0.4817_{(-6.00\%)}$ | $0.5305_{(+3.60\%)}$ | $0.5000_{(-2.40\%)}$ | $0.4939_{(-3.60\%)}$ | $0.4817_{(-6.00\%)}$ | $0.4976_{(-2.90\%)}$ |
| | 20% | $0.4634_{(-9.50\%)}$ | $0.5061_{(-1.20\%)}$ | $0.4939_{(-3.60\%)}$ | $0.4695_{(-8.30\%)}$ | $0.4878_{(-4.80\%)}$ | $0.4841_{(-5.50\%)}$ |
| | 10% | $0.3659_{(-28.60\%)}$ | $0.4634_{(-9.50\%)}$ | $0.4268_{(-16.70\%)}$ | $0.4207_{(-17.90\%)}$ | $0.4451_{(-13.10\%)}$ | $0.4244_{(-17.20\%)}$ |

Table 12: LLaMA-3.1-8B-Instruct on LG-GSM8K: ShotKV vs. Prefill-only.

| Method | 40% | 30% | 20% | 10% |
|---|---|---|---|---|
| ShotKV | 81.07 | 80.82 | 80.57 | 80.37 |
| Prefill-only | 79.07 | 78.82 | 78.57 | 77.26 |

## C.3 MORE EXPERIMENTS ON OTHER MODELS

To further validate the generality of our findings, we also evaluate the impact of KV cache compression on a different model, Mistral-7B-Instruct. As shown in Figure 9, we observe that various KV cache compression methods lead to significant performance degradation across multiple fundamental tasks, especially under aggressive compression ratios. This result demonstrates that the reduction in foundation abilities due to KV cache compression is not limited to a specific model family, but is a general phenomenon affecting different LLM architectures.

Table 9: KV Cache Compression Performance Comparison on *Arithmetic Reasoning* with Different Instruction TuningSettings

| Setting | Ratio | StreamingLLM | H2O | SnapKV | PyramidKV | ChunkKV | Average ↑ |
|---|---|---|---|---|---|---|---|
| | Baseline | | | FullKV: 0.7945 | | | |
| w/ Instruct Tuning | 90% | $0.7695_{(-3.10\%)}$ | $0.7923_{(-0.30\%)}$ | $0.7839_{(-1.30\%)}$ | $0.7854_{(-1.10\%)}$ | $0.7824_{(-1.50\%)}$ | $0.7827_{(-1.50\%)}$ |
| | 80% | $0.7642_{(-3.80\%)}$ | $0.7938_{(-0.10\%)}$ | $0.7824_{(-1.50\%)}$ | $0.7900_{(-0.60\%)}$ | $0.7824_{(-1.50\%)}$ | $0.7826_{(-1.50\%)}$ |
| | 70% | $0.7642_{(-3.80\%)}$ | $0.7900_{(-0.60\%)}$ | $0.7923_{(-0.30\%)}$ | $0.7983_{(+0.50\%)}$ | $0.7809_{(-1.70\%)}$ | $0.7851_{(-1.20\%)}$ |
| | 60% | $0.7650_{(-3.70\%)}$ | $0.7809_{(-1.70\%)}$ | $0.7885_{(-0.80\%)}$ | $0.7923_{(-0.30\%)}$ | $0.7785_{(-0.80\%)}$ | $0.7830_{(-1.50\%)}$ |
| | 50% | $0.7657_{(-3.60\%)}$ | $0.7854_{(-1.10\%)}$ | $0.7847_{(-1.20\%)}$ | $0.7854_{(-1.20\%)}$ | $0.7824_{(-1.50\%)}$ | $0.7807_{(-1.70\%)}$ |
| | 40% | $0.7491_{(-5.70\%)}$ | $0.7688_{(-3.20\%)}$ | $0.7756_{(-2.40\%)}$ | $0.7839_{(-1.30\%)}$ | $0.7763_{(-2.30\%)}$ | $0.7707_{(-3.00\%)}$ |
| | 30% | $0.7051_{(-11.20\%)}$ | $0.7225_{(-9.10\%)}$ | $0.7619_{(-4.10\%)}$ | $0.7718_{(-2.90\%)}$ | $0.7733_{(-2.70\%)}$ | $0.7469_{(-6.00\%)}$ |
| | 20% | $0.6384_{(-19.70\%)}$ | $0.6406_{(-19.40\%)}$ | $0.6884_{(-13.40\%)}$ | $0.7142_{(-10.10\%)}$ | $0.7763_{(-2.30\%)}$ | $0.6916_{(-13.00\%)}$ |
| | 10% | $0.4784_{(-39.80\%)}$ | $0.4503_{(-43.30\%)}$ | $0.5034_{(-36.60\%)}$ | $0.4829_{(-39.20\%)}$ | $0.6566_{(-17.40\%)}$ | $0.5143_{(-35.30\%)}$ |
| | Baseline | | | R1-Distill-Llama-8B FullKV: 0.6938 | | | |
| w/ R1 Distill | 90% | $0.7167_{(+3.30\%)}$ | $0.6900_{(-0.55\%)}$ | $0.6933_{(-0.07\%)}$ | $0.7100_{(+2.34\%)}$ | $0.6867_{(-1.02\%)}$ | $0.6993_{(+0.79\%)}$ |
| | 80% | $0.6867_{(-1.02\%)}$ | $0.6933_{(-0.07\%)}$ | $0.6933_{(-0.07\%)}$ | $0.7067_{(+1.86\%)}$ | $0.6767_{(-2.47\%)}$ | $0.6913_{(-0.36\%)}$ |
| | 70% | $0.6933_{(-0.07\%)}$ | $0.6633_{(-4.40\%)}$ | $0.7100_{(+2.34\%)}$ | $0.7100_{(+2.34\%)}$ | $0.7000_{(+0.89\%)}$ | $0.6953_{(+0.22\%)}$ |
| | 60% | $0.6833_{(-1.51\%)}$ | $0.6900_{(-0.55\%)}$ | $0.6900_{(-0.55\%)}$ | $0.7133_{(+2.81\%)}$ | $0.7067_{(+1.86\%)}$ | $0.6967_{(+0.42\%)}$ |
| | 50% | $0.6700_{(-3.43\%)}$ | $0.6967_{(+0.42\%)}$ | $0.7067_{(+1.86\%)}$ | $0.7000_{(+0.89\%)}$ | $0.6867_{(-1.02\%)}$ | $0.6920_{(-0.26\%)}$ |
| | 40% | $0.6767_{(-2.47\%)}$ | $0.6800_{(-1.99\%)}$ | $0.5967_{(-13.99\%)}$ | $0.6967_{(+0.42\%)}$ | $0.7133_{(+2.81\%)}$ | $0.6727_{(-3.04\%)}$ |
| | 30% | $0.6600_{(-4.87\%)}$ | $0.5900_{(-14.96\%)}$ | $0.5833_{(-15.93\%)}$ | $0.6700_{(-3.43\%)}$ | $0.7000_{(+0.89\%)}$ | $0.6407_{(-7.66\%)}$ |
| | 20% | $0.6200_{(-10.64\%)}$ | $0.4933_{(-28.90\%)}$ | $0.5633_{(-18.81\%)}$ | $0.6833_{(-1.51\%)}$ | $0.6533_{(-5.84\%)}$ | $0.6026_{(-13.14\%)}$ |
| | 10% | $0.5167_{(-25.53\%)}$ | $0.5567_{(-19.76\%)}$ | $0.5767_{(-16.88\%)}$ | $0.6267_{(-9.67\%)}$ | $0.6433_{(-7.28\%)}$ | $0.5840_{(-15.82\%)}$ |
| | Baseline | | | FullKV: 0.5122 | | | |
| w/o Instruct Tuning | 90% | $0.5061_{(-1.20\%)}$ | $0.5183_{(+1.20\%)}$ | $0.5122_{(+0.00\%)}$ | $0.5122_{(+0.00\%)}$ | $0.5122_{(+0.00\%)}$ | $0.5122_{(+0.00\%)}$ |
| | 80% | $0.5061_{(-1.20\%)}$ | $0.5183_{(+1.20\%)}$ | $0.5183_{(+1.20\%)}$ | $0.5305_{(-1.20\%)}$ | $0.5061_{(-1.20\%)}$ | $0.5159_{(+0.70\%)}$ |
| | 70% | $0.5000_{(-2.40\%)}$ | $0.5244_{(+2.40\%)}$ | $0.5122_{(+0.00\%)}$ | $0.5183_{(+1.20\%)}$ | $0.5122_{(+0.00\%)}$ | $0.5134_{(+0.20\%)}$ |
| | 60% | $0.5061_{(-1.20\%)}$ | $0.5366_{(+4.80\%)}$ | $0.5366_{(+4.80\%)}$ | $0.5305_{(+3.60\%)}$ | $0.5244_{(+2.40\%)}$ | $0.5268_{(+2.90\%)}$ |
| | 50% | $0.4939_{(-3.60\%)}$ | $0.5427_{(+6.00\%)}$ | $0.5061_{(-1.20\%)}$ | $0.4939_{(-3.60\%)}$ | $0.4878_{(-4.80\%)}$ | $0.5049_{(-1.40\%)}$ |
| | 40% | $0.4817_{(-6.00\%)}$ | $0.5427_{(+6.00\%)}$ | $0.5244_{(+2.40\%)}$ | $0.4939_{(-3.60\%)}$ | $0.5000_{(-2.40\%)}$ | $0.5085_{(-0.70\%)}$ |
| | 30% | $0.4817_{(-6.00\%)}$ | $0.5305_{(+3.60\%)}$ | $0.5000_{(-2.40\%)}$ | $0.4939_{(-3.60\%)}$ | $0.4817_{(-6.00\%)}$ | $0.4976_{(-2.90\%)}$ |
| | 20% | $0.4634_{(-9.50\%)}$ | $0.5061_{(-1.20\%)}$ | $0.4939_{(-3.60\%)}$ | $0.4695_{(-8.30\%)}$ | $0.4878_{(-4.80\%)}$ | $0.4841_{(-5.50\%)}$ |
| | 10% | $0.3659_{(-28.60\%)}$ | $0.4634_{(-9.50\%)}$ | $0.4268_{(-16.70\%)}$ | $0.4207_{(-17.90\%)}$ | $0.4451_{(-13.10\%)}$ | $0.4244_{(-17.20\%)}$ |

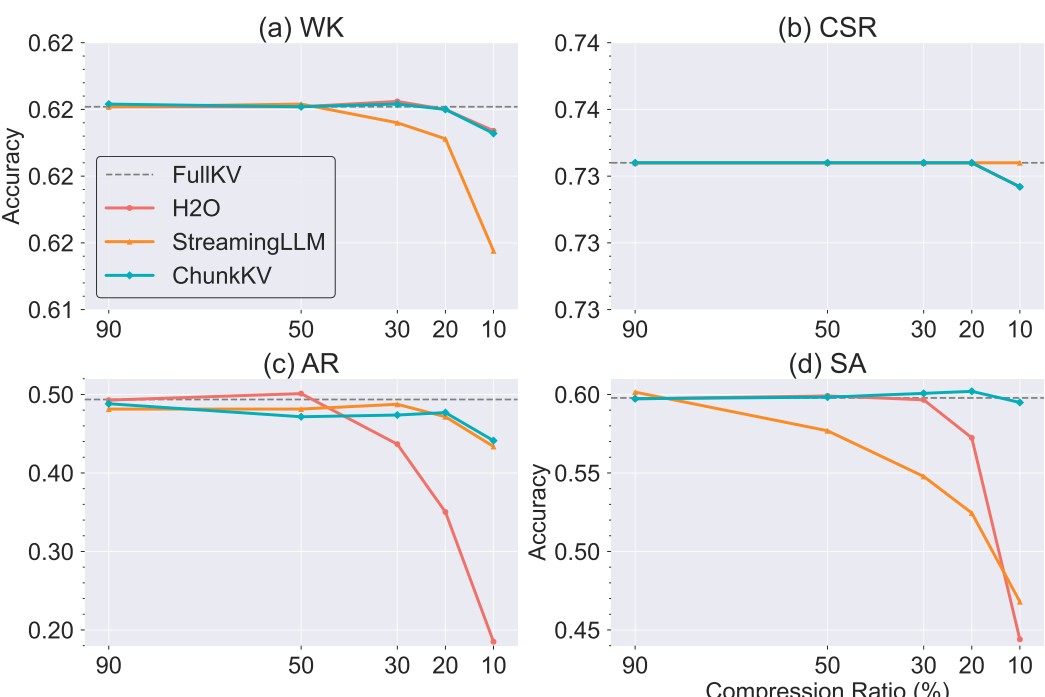

Figure 9: Performance Comparison of KV Cache Compression Methods Across Tasks with Mistral-7B-Instruct.

## D   SHOTKV

This section provides the detailed description of ShotKV.

Table 10: Performance Comparison of Different Shot Numbers on GSM8K

| Shot | Ratio | StreamingLLM | H2O | SnapKV | PyramidKV | ChunkKV | Average ↑ |
|---|---|---|---|---|---|---|---|
| | Baseline | | | FullKV: 0.7149 | | | |
| 1-shot | 90% | $0.7013_{(-1.90\%)}$ | $0.7172_{(+0.30\%)}$ | $0.7142_{(-0.10\%)}$ | $0.7020_{(-1.80\%)}$ | $0.7172_{(+0.30\%)}$ | $0.7104_{(-0.60\%)}$ |
| | 80% | $0.6892_{(-3.60\%)}$ | $0.7089_{(-0.80\%)}$ | $0.7066_{(-1.20\%)}$ | $0.6952_{(-2.80\%)}$ | $0.7081_{(-1.00\%)}$ | $0.7016_{(-1.90\%)}$ |
| | 70% | $0.6816_{(-4.70\%)}$ | $0.6914_{(-3.30\%)}$ | $0.6945_{(-2.90\%)}$ | $0.6884_{(-3.70\%)}$ | $0.7127_{(-0.30\%)}$ | $0.6937_{(-3.00\%)}$ |
| | 60% | $0.6884_{(-3.70\%)}$ | $0.6831_{(-4.40\%)}$ | $0.6914_{(-3.30\%)}$ | $0.6816_{(-4.70\%)}$ | $0.6990_{(-2.20\%)}$ | $0.6887_{(-3.70\%)}$ |
| | 50% | $0.6952_{(-2.80\%)}$ | $0.6596_{(-7.70\%)}$ | $0.6611_{(-7.50\%)}$ | $0.6717_{(-6.00\%)}$ | $0.6732_{(-5.80\%)}$ | $0.6722_{(-6.00\%)}$ |
| | 40% | $0.6657_{(-6.90\%)}$ | $0.6202_{(-13.20\%)}$ | $0.6065_{(-15.20\%)}$ | $0.6475_{(-9.40\%)}$ | $0.6050_{(-15.40\%)}$ | $0.6290_{(-12.00\%)}$ |
| | 30% | $0.5118_{(-28.40\%)}$ | $0.5004_{(-30.00\%)}$ | $0.5042_{(-29.50\%)}$ | $0.5898_{(-17.50\%)}$ | $0.4011_{(-43.90\%)}$ | $0.5015_{(-29.90\%)}$ |
| | 20% | $0.2320_{(-67.50\%)}$ | $0.2714_{(-62.00\%)}$ | $0.2654_{(-62.90\%)}$ | $0.3973_{(-44.40\%)}$ | $0.1319_{(-81.60\%)}$ | $0.2596_{(-63.70\%)}$ |
| | 10% | $0.0296_{(-95.90\%)}$ | $0.0243_{(-96.60\%)}$ | $0.0296_{(-95.90\%)}$ | $0.1236_{(-82.70\%)}$ | $0.0190_{(-97.30\%)}$ | $0.0452_{(-93.70\%)}$ |
| | Baseline | | | FullKV: 0.7574 | | | |
| 2-shot | 90% | $0.7544_{(-0.40\%)}$ | $0.7604_{(+0.40\%)}$ | $0.7574_{(+0.00\%)}$ | $0.7612_{(+0.50\%)}$ | $0.7627_{(+0.70\%)}$ | $0.7592_{(+0.20\%)}$ |
| | 80% | $0.7551_{(-0.30\%)}$ | $0.7521_{(-0.70\%)}$ | $0.7559_{(-0.20\%)}$ | $0.7559_{(-0.20\%)}$ | $0.7589_{(+0.20\%)}$ | $0.7556_{(-0.20\%)}$ |
| | 70% | $0.7521_{(-0.70\%)}$ | $0.7453_{(-1.60\%)}$ | $0.7566_{(-0.10\%)}$ | $0.7574_{(+0.00\%)}$ | $0.7642_{(+0.90\%)}$ | $0.7551_{(-0.30\%)}$ |
| | 60% | $0.7475_{(-1.30\%)}$ | $0.7506_{(-0.90\%)}$ | $0.7521_{(-0.70\%)}$ | $0.7589_{(+0.20\%)}$ | $0.7695_{(+1.60\%)}$ | $0.7557_{(-0.20\%)}$ |
| | 50% | $0.7460_{(-1.50\%)}$ | $0.7437_{(-1.80\%)}$ | $0.7437_{(-1.80\%)}$ | $0.7604_{(+0.40\%)}$ | $0.7619_{(+0.60\%)}$ | $0.7511_{(-0.80\%)}$ |
| | 40% | $0.7445_{(-1.70\%)}$ | $0.7081_{(-6.50\%)}$ | $0.7202_{(-4.90\%)}$ | $0.7309_{(-3.50\%)}$ | $0.7650_{(+1.00\%)}$ | $0.7337_{(-3.10\%)}$ |
| | 30% | $0.7506_{(-0.90\%)}$ | $0.6133_{(-19.00\%)}$ | $0.6657_{(-12.10\%)}$ | $0.7036_{(-7.10\%)}$ | $0.7445_{(-1.70\%)}$ | $0.6955_{(-8.20\%)}$ |
| | 20% | $0.6217_{(-17.90\%)}$ | $0.4412_{(-41.70\%)}$ | $0.4936_{(-34.80\%)}$ | $0.5534_{(-26.90\%)}$ | $0.5368_{(-29.10\%)}$ | $0.5293_{(-30.10\%)}$ |
| | 10% | $0.1516_{(-80.00\%)}$ | $0.1759_{(-76.80\%)}$ | $0.1622_{(-78.60\%)}$ | $0.2244_{(-70.40\%)}$ | $0.0735_{(-90.30\%)}$ | $0.1575_{(-79.20\%)}$ |
| | Baseline | | | FullKV: 0.7597 | | | |
| 4-shot | 90% | $0.7597_{(+0.00\%)}$ | $0.7604_{(+0.10\%)}$ | $0.7650_{(+0.70\%)}$ | $0.7642_{(+0.60\%)}$ | $0.7657_{(+0.80\%)}$ | $0.7630_{(+0.40\%)}$ |
| | 80% | $0.7559_{(-0.50\%)}$ | $0.7688_{(+1.20\%)}$ | $0.7695_{(+1.30\%)}$ | $0.7680_{(+1.10\%)}$ | $0.7642_{(+0.60\%)}$ | $0.7653_{(+0.70\%)}$ |
| | 70% | $0.7597_{(+0.00\%)}$ | $0.7695_{(+1.30\%)}$ | $0.7680_{(+1.10\%)}$ | $0.7710_{(+1.50\%)}$ | $0.7726_{(+1.70\%)}$ | $0.7682_{(+1.10\%)}$ |
| | 60% | $0.7369_{(-3.00\%)}$ | $0.7726_{(+1.70\%)}$ | $0.7688_{(+1.20\%)}$ | $0.7635_{(+0.50\%)}$ | $0.7718_{(+1.60\%)}$ | $0.7627_{(+0.40\%)}$ |
| | 50% | $0.7475_{(-1.60\%)}$ | $0.7612_{(+0.20\%)}$ | $0.7619_{(+0.30\%)}$ | $0.7665_{(+0.90\%)}$ | $0.7635_{(+0.50\%)}$ | $0.7601_{(+0.10\%)}$ |
| | 40% | $0.7165_{(-5.70\%)}$ | $0.7339_{(-3.40\%)}$ | $0.7377_{(-2.90\%)}$ | $0.7483_{(-1.50\%)}$ | $0.7612_{(+0.20\%)}$ | $0.7395_{(-2.70\%)}$ |
| | 30% | $0.6558_{(-13.70\%)}$ | $0.6603_{(-13.10\%)}$ | $0.7111_{(-6.40\%)}$ | $0.7263_{(-4.40\%)}$ | $0.7597_{(+0.00\%)}$ | $0.7026_{(-7.50\%)}$ |
| | 20% | $0.6224_{(-18.10\%)}$ | $0.5625_{(-26.00\%)}$ | $0.6065_{(-20.20\%)}$ | $0.6543_{(-13.90\%)}$ | $0.7468_{(-1.70\%)}$ | $0.6385_{(-16.00\%)}$ |
| | 10% | $0.4708_{(-38.00\%)}$ | $0.3980_{(-47.60\%)}$ | $0.3995_{(-47.40\%)}$ | $0.4321_{(-43.10\%)}$ | $0.3434_{(-54.80\%)}$ | $0.4088_{(-46.20\%)}$ |
| | Baseline | | | FullKV: 0.7680 | | | |
| 6-shot | 90% | $0.7551_{(-1.70\%)}$ | $0.7748_{(+0.90\%)}$ | $0.7839_{(+2.10\%)}$ | $0.7794_{(+1.50\%)}$ | $0.7794_{(+1.50\%)}$ | $0.7745_{(+0.90\%)}$ |
| | 80% | $0.7642_{(-0.50\%)}$ | $0.7756_{(+1.00\%)}$ | $0.7809_{(+1.70\%)}$ | $0.7741_{(+0.80\%)}$ | $0.7786_{(+1.40\%)}$ | $0.7747_{(+0.90\%)}$ |
| | 70% | $0.7513_{(-2.20\%)}$ | $0.7771_{(+1.20\%)}$ | $0.7809_{(+1.70\%)}$ | $0.7771_{(+1.20\%)}$ | $0.7786_{(+1.40\%)}$ | $0.7730_{(+0.70\%)}$ |
| | 60% | $0.7468_{(-2.80\%)}$ | $0.7748_{(+0.90\%)}$ | $0.7733_{(+0.70\%)}$ | $0.7771_{(+1.20\%)}$ | $0.7809_{(+1.70\%)}$ | $0.7706_{(+0.30\%)}$ |
| | 50% | $0.7407_{(-3.60\%)}$ | $0.7718_{(+0.50\%)}$ | $0.7718_{(+0.50\%)}$ | $0.7771_{(+1.20\%)}$ | $0.7718_{(+0.50\%)}$ | $0.7666_{(-0.20\%)}$ |
| | 40% | $0.7377_{(-3.90\%)}$ | $0.7506_{(-2.30\%)}$ | $0.7771_{(+1.20\%)}$ | $0.7688_{(+0.10\%)}$ | $0.7854_{(+2.30\%)}$ | $0.7639_{(-0.50\%)}$ |
| | 30% | $0.7058_{(-8.10\%)}$ | $0.7255_{(-5.50\%)}$ | $0.7392_{(-3.70\%)}$ | $0.7491_{(-2.50\%)}$ | $0.7763_{(+1.10\%)}$ | $0.7392_{(-3.70\%)}$ |
| | 20% | $0.5921_{(-22.90\%)}$ | $0.6232_{(-18.80\%)}$ | $0.6732_{(-12.30\%)}$ | $0.6960_{(-9.40\%)}$ | $0.7665_{(-0.20\%)}$ | $0.6702_{(-12.70\%)}$ |
| | 10% | $0.4572_{(-40.50\%)}$ | $0.4481_{(-41.60\%)}$ | $0.4958_{(-35.40\%)}$ | $0.4458_{(-41.90\%)}$ | $0.5565_{(-27.50\%)}$ | $0.4807_{(-37.40\%)}$ |
| | Baseline | | | FullKV: 0.7945 | | | |
| 8-shot | 90% | $0.7695_{(-3.10\%)}$ | $0.7923_{(-0.30\%)}$ | $0.7839_{(-1.30\%)}$ | $0.7854_{(-1.10\%)}$ | $0.7824_{(-1.50\%)}$ | $0.7827_{(-1.50\%)}$ |
| | 80% | $0.7642_{(-3.80\%)}$ | $0.7938_{(-0.10\%)}$ | $0.7824_{(-1.50\%)}$ | $0.7900_{(-0.60\%)}$ | $0.7824_{(-1.50\%)}$ | $0.7826_{(-1.50\%)}$ |
| | 70% | $0.7642_{(-3.80\%)}$ | $0.7900_{(-0.60\%)}$ | $0.7923_{(-0.30\%)}$ | $0.7983_{(+0.50\%)}$ | $0.7809_{(-1.70\%)}$ | $0.7851_{(-1.20\%)}$ |
| | 60% | $0.7650_{(-3.70\%)}$ | $0.7809_{(-1.70\%)}$ | $0.7885_{(-0.80\%)}$ | $0.7923_{(-0.30\%)}$ | $0.7885_{(-0.80\%)}$ | $0.7830_{(-1.50\%)}$ |
| | 50% | $0.7657_{(-3.60\%)}$ | $0.7854_{(-1.10\%)}$ | $0.7847_{(-1.20\%)}$ | $0.7854_{(-1.10\%)}$ | $0.7824_{(-1.50\%)}$ | $0.7807_{(-1.70\%)}$ |
| | 40% | $0.7491_{(-5.70\%)}$ | $0.7688_{(-3.20\%)}$ | $0.7756_{(-2.40\%)}$ | $0.7839_{(-1.30\%)}$ | $0.7763_{(-2.30\%)}$ | $0.7707_{(-3.00\%)}$ |
| | 30% | $0.7051_{(-11.20\%)}$ | $0.7225_{(-9.10\%)}$ | $0.7619_{(-4.10\%)}$ | $0.7718_{(-2.90\%)}$ | $0.7733_{(-2.70\%)}$ | $0.7469_{(-6.00\%)}$ |
| | 20% | $0.6384_{(-19.70\%)}$ | $0.6406_{(-19.40\%)}$ | $0.6884_{(-13.40\%)}$ | $0.7142_{(-10.10\%)}$ | $0.7763_{(-2.30\%)}$ | $0.6916_{(-13.00\%)}$ |
| | 10% | $0.4784_{(-39.80\%)}$ | $0.4503_{(-43.30\%)}$ | $0.5034_{(-36.60\%)}$ | $0.4829_{(-39.20\%)}$ | $0.6566_{(-17.40\%)}$ | $0.5143_{(-35.30\%)}$ |

## D.1 PSEUDOCODE

The detailed algorithm of ShotKV is presented in Algorithm1. Our method consists of two main phases: prefill compression and decoding compression. During the prefill phase, we compute an importance score for each shot by averaging the attention weights across all tokens, heads, and layers within that shot. This score $\text{Score}_{\text{prefill}}(s_i)$ is normalized by the shot length $k_i$ to avoid bias towards longer shots. Shots are then sorted by their scores and preserved until reaching the specified prefill ratio $r_p$.

In the decoding phase, compression is performed dynamically at each step. For each token in the decoding KV cache, we calculate its importance score $\text{Score}_{\text{decoding}}(t)$ by summing attention weights across all heads and layers. The top-k tokens are retained based on the decoding ratio $r_d$. Finally, the compressed KV cache is formed by combining both the preserved prefill and decoding caches.

Table 11: Performance Comparison of Different KV Cache Compression Methods on Many-shot GSM8K

| Benchmark | Ratio | StreamingLLM | H2O | SnapKV | PyramidKV | ChunkKV | Average ↑ |
|---|---|---|---|---|---|---|---|
| | Baseline | | LLaMA-3.1-8B-Instruct FullKV: 0.8235 | | | | |
| | 90% | $0.7728_{(-6.16\%)}$ | $0.8142_{(-1.13\%)}$ | $0.8137_{(-1.19\%)}$ | $0.7932_{(-3.68\%)}$ | $0.8233_{(-0.02\%)}$ | $0.8034_{(-2.44\%)}$ |
| | 80% | $0.7935_{(-3.64\%)}$ | $0.8334_{(+1.20\%)}$ | $0.8138_{(-1.18\%)}$ | $0.8037_{(-2.40\%)}$ | $0.7932_{(-3.68\%)}$ | $0.8075_{(-1.94\%)}$ |
| | 70% | $0.8038_{(-2.39\%)}$ | $0.8136_{(-1.20\%)}$ | $0.7832_{(-4.89\%)}$ | $0.7932_{(-3.68\%)}$ | $0.8037_{(-2.40\%)}$ | $0.7995_{(-2.91\%)}$ |
| | 60% | $0.7932_{(-3.68\%)}$ | $0.8142_{(-1.13\%)}$ | $0.8037_{(-2.40\%)}$ | $0.7935_{(-3.64\%)}$ | $0.8038_{(-2.39\%)}$ | $0.8017_{(-2.65\%)}$ |
| | 50% | $0.7934_{(-3.65\%)}$ | $0.8137_{(-1.19\%)}$ | $0.7932_{(-3.68\%)}$ | $0.7932_{(-3.68\%)}$ | $0.7835_{(-4.86\%)}$ | $0.7954_{(-3.41\%)}$ |
| | 40% | $0.8037_{(-2.40\%)}$ | $0.7832_{(-4.89\%)}$ | $0.7935_{(-3.64\%)}$ | $0.7834_{(-4.87\%)}$ | $0.7832_{(-4.89\%)}$ | $0.7894_{(-4.14\%)}$ |
| | 30% | $0.7835_{(-4.86\%)}$ | $0.7932_{(-3.68\%)}$ | $0.8038_{(-2.39\%)}$ | $0.7934_{(-3.65\%)}$ | $0.7932_{(-3.68\%)}$ | $0.7934_{(-3.65\%)}$ |
| Many-shot | 20% | $0.7537_{(-8.47\%)}$ | $0.7428_{(-9.80\%)}$ | $0.7934_{(-3.65\%)}$ | $0.7832_{(-4.89\%)}$ | $0.7835_{(-4.86\%)}$ | $0.7713_{(-6.34\%)}$ |
| GSM8K | 10% | $0.7432_{(-9.75\%)}$ | $0.5127_{(-37.74\%)}$ | $0.6827_{(-17.10\%)}$ | $0.7037_{(-14.55\%)}$ | $0.7932_{(-3.68\%)}$ | $0.6871_{(-16.56\%)}$ |
| | Baseline | | R1-Distill-Llama-8B FullKV: 0.7123 | | | | |
| | 90% | $0.7123_{(+1.42\%)}$ | $0.6612_{(-5.85\%)}$ | $0.6534_{(-6.96\%)}$ | $0.6912_{(-1.58\%)}$ | $0.6923_{(-1.42\%)}$ | $0.6821_{(-2.88\%)}$ |
| | 80% | $0.7234_{(+3.00\%)}$ | $0.6534_{(-6.96\%)}$ | $0.7123_{(+1.42\%)}$ | $0.6423_{(-8.54\%)}$ | $0.7123_{(+1.42\%)}$ | $0.6887_{(-1.94\%)}$ |
| | 70% | $0.7412_{(+5.54\%)}$ | $0.6523_{(-7.12\%)}$ | $0.7234_{(+3.00\%)}$ | $0.6923_{(-1.42\%)}$ | $0.7234_{(+3.00\%)}$ | $0.7065_{(+0.60\%)}$ |
| | 60% | $0.7423_{(+5.69\%)}$ | $0.6912_{(-1.58\%)}$ | $0.6912_{(-1.58\%)}$ | $0.6823_{(-2.85\%)}$ | $0.6634_{(-5.54\%)}$ | $0.6941_{(-1.17\%)}$ |
| | 50% | $0.7234_{(+3.00\%)}$ | $0.7134_{(+1.58\%)}$ | $0.7312_{(+4.12\%)}$ | $0.7123_{(+1.42\%)}$ | $0.7123_{(+1.42\%)}$ | $0.7185_{(+2.31\%)}$ |
| | 40% | $0.7123_{(+1.42\%)}$ | $0.6923_{(-1.42\%)}$ | $0.6923_{(-1.42\%)}$ | $0.7023_{(+0.00\%)}$ | $0.7234_{(+3.00\%)}$ | $0.7045_{(+0.31\%)}$ |
| | 30% | $0.6523_{(-7.12\%)}$ | $0.7312_{(+4.12\%)}$ | $0.6634_{(-5.54\%)}$ | $0.7423_{(+5.69\%)}$ | $0.6912_{(-1.58\%)}$ | $0.6961_{(-0.88\%)}$ |
| | 20% | $0.6912_{(-1.58\%)}$ | $0.5834_{(-16.93\%)}$ | $0.5123_{(-27.05\%)}$ | $0.6823_{(-2.85\%)}$ | $0.6634_{(-5.54\%)}$ | $0.6265_{(-10.79\%)}$ |
| | 10% | $0.6323_{(-9.97\%)}$ | $0.5423_{(-22.78\%)}$ | $0.5412_{(-22.94\%)}$ | $0.5923_{(-15.66\%)}$ | $0.6823_{(-2.85\%)}$ | $0.5981_{(-14.84\%)}$ |

This two-phase approach allows for different compression strategies during prefill and decoding stages, recognizing their distinct roles in the inference process. The shot-aware design during prefill ensures that the most informative examples are preserved, while the token-level compression during decoding maintains essential recent context.

---

**Algorithm 1** ShotKV: Shot-aware KV Cache Compression

---

**Require:** Prompt with $n$ shots $\{s_1, ..., s_n\}$, prefill ratio $r_p$, decoding ratio $r_d$
**Ensure:** Compressed KV cache $KV_{\text{total}}$
1: // Phase 1: Prefill Compression (performed once)
2: **for** each shot $s_i$ in $\{s_1, ..., s_n\}$ **do**
3:      Compute $\text{Score}_{\text{prefill}}(s_i) = \frac{1}{k_i} \sum_{t \in s_i} \sum_{h=1}^{H} \sum_{l=1}^{L} \alpha_{t,h}^{l}$
4: **end for**
5: Sort shots by $\text{Score}_{\text{prefill}}(s_i)$ in descending order
6: $S_{\text{preserved}} \leftarrow$ Select shots until $\sum_{s_i} k_i \leq r_p \times |KV_{\text{prefill}}|$
7: $KV_{\text{prefill}}^{C} \leftarrow \text{Compress}(\{s_i | s_i \in S_{\text{preserved}}\})$
8: // Phase 2: Decoding Compression (performed dynamically)
9: **for** each decoding step **do**
10:      **for** each token $t$ in $KV_{\text{decoding}}$ **do**
11:          Compute $\text{Score}_{\text{decoding}}(t) = \sum_{h=1}^{H} \sum_{l=1}^{L} \alpha_{t,h}^{l}$
12:      **end for**
13:      $k \leftarrow r_d \times |KV_{\text{decoding}}|$
14:      $KV_{\text{decoding}}^{C} \leftarrow \text{TopK}(KV_{\text{decoding}}, \text{Score}_{\text{decoding}}, k)$
15: **end for**
       **return** $KV_{\text{prefill}}^{C} \cup KV_{\text{decoding}}^{C}$

---

# E EVALUATION BENCHMARK

## E.1 DATASET DETAILS

Detailed statistics for each benchmark dataset are provided in Table 13. For HotpotQA, we only report results under the 10% compression ratio using the LLaMA-3-8B-Instruct model.

| DATASET | TASK TYPE | # TEST | METRIC | EVALUATION METHOD |
|---|---|---|---|---|
| MMLU Hendrycks et al. (2020) | World Knowledge | 14,079 | Accuracy | Generation-Based |
| GSM8K Cobbe et al. (2021) | Arithmetic | 1,319 | Exact match | Generation-Based |
| CSQA Talmor et al. (2019) | Commonsense | 1,221 | Accuracy | Generation-Based |
| HumanEval Chen et al. (2021) | Code Generation | 164 | Pass@1 rate | Generation-Based |
| JailBreakV Luo et al. (2024) | Safety | 28,000 | Attack success rate | Generation-Based |
| HotpotQA Yang et al. (2018) | Document QA (Multi-hop) | 7,405 | Accuracy | Generation-Based |
| LongGenBench Liu et al. (2024d) | Long-Context Generation | 23,000 | Accuracy | Generation-Based |

Table 13: The statistics of the datasets used in this paper. # TEST denote the number of training data and test data, respectively.

## F    IMPACT STATEMENT

This work advances the field of efficient large language model deployment through systematic analysis and improvement of KV cache compression techniques. Our research has several potential societal impacts:

First, by enabling more efficient memory usage in LLMs while maintaining performance, our work contributes to reducing the computational resources and energy consumption required for AI deployment. This has positive environmental implications and makes AI technology more accessible to researchers and organizations with limited computing resources.

Second, our proposed ShotKV method specifically improves performance on long-context arithmetic reasoning tasks, which could enhance the practical applications of LLMs in education, scientific computing, and other fields requiring complex mathematical reasoning. This could lead to more reliable AI-assisted learning and problem-solving tools.

However, we acknowledge that making LLMs more efficient could accelerate their widespread adoption, potentially raising concerns about AI's impact on employment and privacy. While our work focuses on technical improvements, we encourage the research community to carefully consider these broader implications when deploying such technologies.

We believe the benefits of more efficient and capable AI systems outweigh potential risks, particularly as our work promotes more sustainable and accessible AI development. Nevertheless, we emphasize the importance of responsible deployment and continued ethical consideration in the application of these technologies.

