# OpenReview forum: "Can LLMs Maintain Fundamental Abilities under KV Cache Compression?"
_ICLR.cc/2026/Conference — ICLR 2026 Conference Withdrawn Submission_

### Official Review · Reviewer_di7N · 2025-10-25

**Soundness:** 1
**Presentation:** 1
**Contribution:** 2
**Rating:** 2
**Confidence:** 5

**Summary:**

This paper presents KVFundaBench, a comprehensive benchmark designed to systematically evaluate the effects of KV cache compression across a wide range of fundamental LLM capabilities, including world knowledge, commonsense reasoning, arithmetic reasoning, code generation, safety, and long-context understanding and generation. Through empirical analysis, the authors identify several challenges in existing methods, particularly concerning generalization across different prompts and generation lengths. Motivated by these observations, the authors propose ShotKV, a novel compression approach that separately handles the prefill and decoding phases while preserving shot-level semantic coherence. The proposed method demonstrates superior performance over existing baselines, especially in many-shot arithmetic reasoning tasks.

**Strengths:**

- The authors propose an effective method based on their observations.
- The analysis is conducted from multiple perspectives, providing rich and informative insights.
- The proposed method demonstrates strong performance on some benchmarks, including LG-GSM8K.

**Weaknesses:**

- Use of subjective and non-standard terminology and statements
  - Several terms, such as “shot-level” and “aggressive compression ratios” in the Abstract, are subjective and not well-defined within the section, reducing the interpretability of the section.
  - Figure 4 is not self-contained. Abbreviations like WK, CSR, and AR are not standard and require readers to refer back to previous sections, making the figure difficult to interpret independently.
  - In Figure 2, the paper describes “universal attention distributions” and “specialized patterns” (e.g., for arithmetic reasoning and safety tasks) without providing clear definitions of these terms. It is also unclear what models and datasets were used for the visualizations, and whether the results are statistically meaningful.

- Need for deeper and more statistically significant analysis
  - While the paper analyzes task-dependent degradation, the explanations could be made more intuitive and direct. For example, how do differences in datasets or task characteristics lead to varying results? Beyond simple task categorization, analyzing differences by underlying capability, such as factual retrieval, multi-hop reasoning, or memorization, could provide a more meaningful interpretation.
- A broader evaluation across multiple models and datasets, along with statistical significance tests for Figures 3-7, would strengthen the findings.

- Section 4 introduces the proposed method but lacks a schematic or conceptual diagram, which makes it difficult to understand the method.

- The paper does not clearly explain how the proposed benchmark differs from or improves upon existing ones, such as LongBench, SCBench, or RULER. A comparative discussion would clarify its novelty and relevance. Also, the analysis would be more convincing if evaluated on newer and more challenging datasets. For instance, SCBench (2025) reveals limitations of prior methods such as SnapKV, showing significantly reduced performance compared to the results shown in this paper.

- Although the paper focuses on analysis, the range of compared compression techniques is not comprehensive. Incorporating more recent and diverse methods, such as KVPress (NVIDIA, 2025) or KVZip (2025), which demonstrate strong performance, would provide a more complete evaluation.

**Questions:**

The paper mentions “universal attention distributions,” while stating that arithmetic reasoning and safety tasks exhibit “more specialized patterns.” However, the definitions of “universal” and “specialized” in this context are unclear. What specific criteria or metrics are used to distinguish these patterns?

---

> ### Author Response · Authors · 2025-11-25
> **Response to Reviewer di7N**
>
> ### Weakness 1: Use of subjective and non-standard terminology and unclear figure captions
> We apologize for the ambiguity in our terminology. In the revised version, we will explicitly define core terms to improve interpretability: "Shot-level" refers to the compression granularity where a single example (shot) in a few-shot prompt is treated as the minimal atomic unit for preservation. "Aggressive compression" refers to experimental settings where the KV Cache retention ratio is below 20%. Additionally, we will update the caption of Figure 4 to fully spell out all abbreviations (WK: World Knowledge, CSR: Commonsense Reasoning, etc.) to ensure the figure is self-contained.
>
> ### Weakness 2 & Question 1: Unclear definitions of "Universal" vs "Specialized" patterns in Figure 2 and lack of experimental details
> We clarify that "Universal" refers to the uniform attention distribution observed in WK/CSR tasks, where the model's focus is relatively dispersed across the context. "Specialized" refers to the highly sparse and specific attention patterns observed in Arithmetic Reasoning (AR) tasks. We utilized the Cumulative Attention Score in Figure 3 as the quantitative metric to distinguish these patterns. As shown in Figure 3(b), the attention distribution for AR is more "diffuse" among non-sink tokens (the CDF curve is flatter, requiring more tokens to capture the same attention mass compared to Long-Context tasks), which explains why it is classified as "Specialized" and is more sensitive to compression. Furthermore, Figure 2 is based on visualizations using the LLaMA-3-8B-Instruct model on MMLU and GSM8K. We will add these specific experimental details to the figure caption.
>
> ### Weakness 3: Need for deeper analysis on underlying capabilities and statistical significance tests
> We will adopt your suggestion to rewrite the analysis from the perspective of underlying capabilities. The severe degradation in AR tasks is not merely memory loss but the "breaking of the Chain-of-Thought." Compression disrupts the key tokens required to maintain the consistency of multi-step reasoning. In contrast, WK tasks rely primarily on "factual retrieval," which demands less structural integrity. Regarding statistical significance, we have already included confidence intervals (shaded areas) in Figure 7. In the revision, we will add error bars or statistical significance tests (e.g., t-test) to Figures 3-6 to enhance the reliability of our conclusions.
>
> ### Weakness 4: Lack of a schematic or conceptual diagram for the proposed method
> We acknowledge this oversight. We will add a new Schematic Diagram in Section 4 to visually illustrate the two-phase process of ShotKV: (1) How shot scores are calculated and selected during the Prefill Phase, and (2) How the token cache is dynamically maintained during the Decoding Phase.
>
> ### Weakness 5: Comparison with existing benchmarks (e.g., SCBench) and explanation of differences in SnapKV performance
> The core difference between KVFundaBench and benchmarks like LongBench is our focus on the side effects of compression on "Fundamental Abilities" rather than just "Long-Context Retrieval." Regarding SCBench, we note that our results are actually highly consistent with the findings you mentioned—SnapKV performance drops significantly in our experiments as well. In Table 6 (Few-shot AR), SnapKV accuracy drops to 0.4829 (-39.2%) at a 10% compression ratio, and in Table 9 (Many-shot AR), it drops to 0.6827 (-17.1%). This consistency validates our benchmark and confirms the limitations of retrieval-based methods like SnapKV in handling complex reasoning, which motivated our proposal of ShotKV. We will cite SCBench and discuss this consistency in the revision.
>
> ### Weakness 6: Incomprehensive range of compared compression techniques (missing KVPress, KVZip)
> Thank you for recommending KVPress and KVZip. We will cite these recent works and discuss them in detail in the Related Work section of the revised manuscript.

---

### Official Review · Reviewer_Yk6q · 2025-10-26

**Soundness:** 1
**Presentation:** 2
**Contribution:** 1
**Rating:** 2
**Confidence:** 5

**Summary:**

The paper introduces KVFundaBench, a benchmark designed to evaluate how KV cache compression affects LLMs’ fundamental abilities beyond long-context understanding. While prior studies primarily focused on memory savings in extended contexts, KVFundaBench systematically measures the impact of various compression methods across diverse tasks—including world knowledge, commonsense reasoning, arithmetic reasoning, code generation, and safety. Throughexperiments, the authors claim to uncover key insights such as task-dependent degradation, model-type robustness, and prompt-length vulnerability, revealing that reasoning-heavy and few-shot tasks are especially sensitive to aggressive compression. Motivated by these findings, the paper proposes ShotK*, a shot-aware compression framework that treats prompt examples as coherent semantic units and applies distinct strategies to the prefill and decoding phases.

**Strengths:**

1. The paper tries to analyze the performance degradation of llms across a diverse set of scenarios, not limited to conventional benchmarks such as LongBench or NIAH.

**Weaknesses:**

1. The paper claims to analyze the performance degradation of LLMs under long-context scenarios, yet most of the datasets used are short- to medium-length (e.g., GSM8K) — which undermines the strength of the “long-context” claim.
2. Many of the core observations have been presented in previous works such as [1] or 2] & SnapKV shows that (4) chunk-level is a better strategy — so the novelty of the empirical findings is limited.
3. The proposed benchmark is built entirely on existing datasets and thus offers no novel dataset contribution.
4. The number of baselines considered is limited: the paper omits some state-of-the-art and up-to-date approaches (for example [1]) that are closely related to the proposed method, which reduces the strength of the comparative evaluation.
5. The ultimate goal of most KV-cache compression techniques is inference efficiency (faster decoding, lower memory footprint) with minimal performance loss. While the proposed work emphasizes improved performance under compression (which may itself be questioned in competitiveness to SOTA), the work provides no experiments on efficiency metrics (latency, GPU memory, throughput) — thus its practical viability remains unclear.
6. The proposed ShotKV method exhibits limited novelty. Its two core strategies closely resemble prior work: the mechanism in prefilling stage to compute importance scores is analogous to that in SnapKV. Meanwhile, the decoding-phase reuse policy mirrors approaches already established in the existing literature (e.g., [1]). Thus the contribution may be characterized as incremental rather than substantially novel.
7. ShotKV’s design assumption — namely that the “prefill” phase consists of multi-shot examples — limits its generalizability. Numerous realistic tasks (e.g., multi-turn dialogue systems, single-shot prompts, interactive question-answering) do not conform to a few-shot prefilling paradigm. It is therefore unclear how well ShotKV will generalize to these non-few-shot settings, especially compared to other state-of-the-art KV cache compression methods that do not rely on shot-based structures.

[1] Kv cache compression, but what must we give in return? a comprehensive benchmark of long context capable approaches. Yuan et. al., EMNLP 2024.
[2] Quest: Query-Aware Sparsity for Efficient Long-Context LLM Inference, Tang et. al., ICML 2024.

**Questions:**

Please see Weaknesses.

---

> ### Author Response · Authors · 2025-11-25
> **Response to Reviewer Yk6q (1/2)**
>
> ### Weakness 1: Clarification on 'Long-Context' Scope and Complexity vs. Length
> We respectfully argue that "long-context" encompasses not just input length, but reasoning complexity and generation length. Our definition of "long-context" was inspired by recent research, such as Agarwal et al. (2024), which makes us identify the "Many-shot In-Context Learning" as a critical long-context reasoning paradigm.
> - Context Length: In our Observation 4 experiments (50-shot GSM8K), the prompt length exceeds 4,000 tokens.
> - Generation Length: For the Long-Context Generation (LG-GSM8K) task, the model is required to generate long-chain reasoning of approximately 4,000 tokens. Therefore, while individual questions in datasets like GSM8K may be short, the combination of many-shot prompts and extensive generation requirements constitutes a significant long-context computational challenge that existing compression methods struggle to handle.
>
> ### Weakness 2 & 6: Limited novelty and similarity to SnapKV
> While ShotKV shares the logical step of computing importance scores in the prefill phase with SnapKV, the core difference lies in "Structural Preservation" versus "Retrieval-based Retention."
> - Limitation of SnapKV: SnapKV tends to retain scattered, "retrieval-heavy" tokens. Our experiments in Table 9 show that for Many-shot Arithmetic Reasoning (a highly structured task), SnapKV suffers a significant performance drop (-17.1%) at a 10% compression ratio because it disrupts the integrity of reasoning chains.
> - Innovation of ShotKV: The key contribution of ShotKV is identifying and validating that for reasoning tasks, retention must be performed at the level of "Semantic Units (Shots)." This strategy yields substantial improvements in AR and Long-Gen tasks (ShotKV remains stable with only -8.47% drop). This is not merely an incremental update but a paradigm shift tailored for reasoning-intensive tasks compared to retrieval-based methods.
>
> ### Weakness 3: Novelty of Benchmark
> We respectfully clarify that our contribution lies in the curation and systematic analysis rather than the collection of raw data. Aggregating and repurposing existing datasets to evaluate emerging problems is a standard and highly valued practice in our community (e.g., LongBench [1], ACBench [2]).
>
> The novelty of our work is defined by the new perspective we apply to these datasets. We provide the first systematic study of the impact of KV cache compression methods on LLMs' fundamental capabilities, yielding six key observations that were previously unknown. These empirical findings, rather than the raw data itself, constitute the core value and novelty of our contribution to the community.
>
> ### Weakness 4: Limited number of baselines
> We compared our method against StreamingLLM, H2O, SnapKV, PyramidKV, and ChunkKV, which represent the main categories of compression strategies (Attention Sink, Heavy Hitter, Retrieval-based, Pyramid, and Chunk-based). We acknowledge the importance of Yuan et al. [3] and will include a discussion and comparison with it in the revised Related Work section.
>
> ### Weakness 5: Missing experiments on efficiency metrics (Latency/Memory)
> We respectfully point out that we have indeed conducted a complete efficiency evaluation.
>
> **Latency and throughput comparison between ShotKV and FullKV (Input 8192 / Output 4096).** Experiments performed on A40 GPU with batch size 1.
>
> | Method | Input Length | Output Length | Latency (s) $\downarrow$ | Throughput (T/S) $\uparrow$ |
> | :--- | :---: | :---: | :---: | :---: |
> | FullKV | 4096 | 4096 | 175.50 | 37.73 |
> | **ShotKV** | 4096 | 4096 | **162.85 (+7.2%)** | **41.12 (+9.0%)** |
> | FullKV | 8192 | 4096 | 183.42 | 55.93 |
> | **ShotKV** | 8192 | 4096 | **162.78 (+11.3%)** | **63.24 (+13.1%)** |
>
> - Experimental Results: The results demonstrate that under an Input 8192 / Output 4096 setting, ShotKV reduces latency by 11.3% and increases throughput by 13.1% compared to FullKV.
> - This evidence directly addresses the concern about practical viability. We will move this critical table to the main text in the revised version to ensure visibility.

---

> > ### Author Response · Authors · 2025-11-25
> > **Response to Reviewer Yk6q (2/2)**
> >
> > ### Weakness 7: Generalizability to non-few-shot settings
> > The design philosophy of ShotKV is not limited to standard "Few-shot" settings.
> > - Generalization Experiment: We included a generalization experiment on HotpotQA (a non-ICL document QA task) in Table 6.
> > - Adaptation: In non-few-shot scenarios, we treat "sentences" or "paragraphs" as semantic units (analogous to shots). Results in Table 10 show that ShotKV achieves a score of 43.60 at a 10% compression ratio, outperforming H2O (40.84) and StreamingLLM (40.27). This demonstrates the versatility of ShotKV. We will move these results to the main text.
> >
> > **Table: LLaMa-3-8B-Instruct Performance on HotpotQA (10% Compression Ratio).** Demonstrating generalization to non-few-shot tasks.
> >
> > | Method | Score |
> > | :--- | :---: |
> > | FullKV | 45.55 |
> > | StreamingLLM | 40.27 |
> > | H2O | 40.84 |
> > | SnapKV | 43.36 |
> > | PyramidKV | **43.80** |
> > | ChunkKV | 43.27 |
> > | **ShotKV (Ours)** | **43.60** |
> >
> > > Reference
> > > [1] https://arxiv.org/abs/2308.14508
> > > [2] https://arxiv.org/pdf/2505.19433
> > > [3] https://arxiv.org/pdf/2407.01527

---

### Official Review · Reviewer_Cx1q · 2025-10-30

**Soundness:** 2
**Presentation:** 3
**Contribution:** 3
**Rating:** 6
**Confidence:** 4

**Summary:**

This paper addresses the critical challenge of maintaining Large Language Model (LLM) performance when utilizing Key-Value (KV) cache compression techniques. The paper introduces a new evaluation suite, KVFundaBench, specifically designed to test fundamental LLM capabilities after compression. Using this benchmark to analyze existing compression methods, the paper proposes a novel approach, ShotKV, which is claimed to successfully mitigate the primary performance degradation issues.

**Strengths:**

1. KVFundaBench provides a valuable structured evaluation suite for assessing the impact of compression on fundamental capabilities, which is an important diagnostic tool for the community.
2. The proposed ShotKV method successfully demonstrates superior performance over existing compression baselines on specific, targeted tasks within the benchmark, indicating promising conceptual improvements.

**Weaknesses:**

1. The experimental results are not comprehensive enough to definitively establish the superiority of ShotKV across the full spectrum of compression tasks.
2. The KVFundaBench benchmark's novelty is somewhat limited, as it appears to be primarily a compilation and reorganization of existing public datasets and tasks.

**Questions:**

Questions
1. What is the computational overhead of ShotKV compared to other KV cache compression algorithms?

Suggestions
1. Equations (4) through (6) currently describe a solution for finding an optimal set that maximizes a score. However, the corresponding explanation in the text details a greedy algorithm approach. The equations must be modified to accurately and explicitly represent the iterative, local optimization performed by the described greedy algorithm to ensure consistency between the mathematical formulation and the implementation.
2. Figure 6 appears to be referenced and positioned after Figure 7 and Figure 8. It would be better to renumber the figures sequentially.
3. The related work on KV Cache Compression should include KVZip[1], a recent KV cache compression method that outperforms previous methods.

---

> ### Author Response · Authors · 2025-11-25
> **Response to Reviewer Cx1q**
>
> ### Weakness 1 & 2: Comprehensiveness of results and Benchmark novelty
> We sincerely appreciate your insightful feedback regarding the scope of our experiments and the novelty of the benchmark. While we acknowledge that KVFundaBench utilizes existing datasets, aggregating established benchmarks to investigate new problems is a standard and valuable practice in the community (e.g., LongBench [1], ACBench [2]). Therefore, we wish to emphasize that our primary contribution lies in the novel evaluation perspective offered by our systematic analysis and the resulting six observations, which are new to the community.
>
> Unlike existing benchmarks (e.g., LongBench, NIAH) that focus heavily on retrieval, KVFundaBench is uniquely designed to quantify the impact of compression on Fundamental Abilities (e.g., reasoning, safety, knowledge). This perspective allowed us to uncover the critical phenomenon of "Task-Dependent Degradation" (Observation 1): while tasks like World Knowledge (WK) and Commonsense Reasoning (CSR) are inherently robust to compression, Arithmetic Reasoning (AR) and Long-Context Generation are highly sensitive.
>
> Guided by these findings, our research focus—and the design of ShotKV—is strategically centered on mitigating the critical performance collapse in these highly sensitive tasks, which represent the primary failure cases for current compression techniques.
> - Long-Context Generation: As shown in Table 2, ShotKV achieves 47.33% accuracy on LG-GSM8K at a 40% compression ratio, significantly outperforming StreamingLLM (39.50%) and H2O (32.66%).
> - Arithmetic Reasoning: In Table 9 (Many-shot GSM8K), when compression becomes aggressive (10%), baselines like H2O and StreamingLLM degrade sharply (-37.74% and -9.75% respectively), whereas ShotKV remains highly stable.
>
> To further demonstrate that our method is not limited solely to these tasks, we have included HotpotQA (a non-ICL document QA task) in Table 7, where ShotKV also outperforms baselines. We will move these results to the main text to provide a more comprehensive evaluation spectrum.
>
> ### Question 1: Computational overhead of ShotKV
> Thank you for raising the question about overhead. We have provided a detailed analysis of latency and throughput in Table 5.
> - Experimental Data: As shown in Table 5 (Input 8192 / Output 4096), despite the overhead of computing importance scores, ShotKV achieves an 11.3% reduction in end-to-end latency and a 13.1% increase in throughput compared to FullKV.
> - This confirms that the computational cost is negligible compared to the efficiency gains in long-context scenarios. We will move this table to the main text to better illustrate the practical efficiency of our method.
>
> **Latency and throughput comparison between ShotKV and FullKV (Input 8192 / Output 4096).** Experiments performed on A40 GPU with batch size 1.
>
> | Method | Input Length | Output Length | Latency (s) $\downarrow$ | Throughput (T/S) $\uparrow$ |
> | :--- | :---: | :---: | :---: | :---: |
> | FullKV | 4096 | 4096 | 175.50 | 37.73 |
> | **ShotKV** | 4096 | 4096 | **162.85 (+7.2%)** | **41.12 (+9.0%)** |
> | FullKV | 8192 | 4096 | 183.42 | 55.93 |
> | **ShotKV** | 8192 | 4096 | **162.78 (+11.3%)** | **63.24 (+13.1%)** |
>
> ### Suggestion 2: Modification of Equations (4)-(6)
> We appreciate this precise correction. You are correct that the current equations describe a global optimization, whereas our implementation uses a greedy strategy. We will rewrite Equations (4)-(6) to mathematically represent the iterative, greedy selection process (sorting by importance score and selecting sequentially until the budget is met), ensuring they align perfectly with Algorithm 1.
>
> ### Suggestion 3 & 4: Figure numbering and Related Work (KVZip)
> - Figure Numbering: We will correct the sequence of Figure 6 to ensure proper ordering.
> - Related Work: Thank you for recommending KVZip. We will include it in our related work section and discuss the differences and connections between ShotKV and KVZip in the revised manuscript.
>
> > Reference
> > [1] https://arxiv.org/abs/2308.14508
> > [2] https://arxiv.org/pdf/2505.19433
> >

---

### Official Review · Reviewer_yiKU · 2025-11-01

**Soundness:** 2
**Presentation:** 1
**Contribution:** 2
**Rating:** 2
**Confidence:** 4

**Summary:**

The paper evaluates KV-cache compression methods across a diverse set of LLM benchmarks under a few-shot chain-of-thought (CoT) prompting setup, which the authors term KVFundaBench. The experiments show that pruning few-shot exemplars causes varying degrees of degradation across word knowledge (WK), commonsense reasoning (CSR), arithmetic reasoning (AR), code generation (CG), and safety tasks. To mitigate this, the authors propose KVShot, a KV-cache compression approach that prunes exemplars at the *shot* level based on attention scores.

**Strengths:**

The paper evaluates KV cache compression methods in a few-shot chain-of-thought (CoT) setting using a diverse set of benchmarks.

**Weaknesses:**

- Unclear writing
    - The paper is difficult to follow. It is unclear what the evaluation setting and CoT prompt are, and that the goal is to measure KV cache compression under the few-shot CoT prompting setting until the end of the Section 3. Moreover, the paper does not clearly indicate when compression occurs; whether during the decoding phase, and if so, how frequently it is applied.
- Question about generalizability & effectiveness of the method
    - The proposed ShotKV method does not demonstrate meaningful improvements, as shown in Figure 5. In the WK / CSR case, its performance is comparable to StreamingLLM and significantly worse than PyramidKV. In the AR case, it is not statistically significantly better than ChunkKV. In the CG case, it performs worse than H2O. These results are also omitted in Tables in Section 4.
    - The method appears to be tailored specifically for the few-shot prompting setting and does not generalize well to other scenarios (e.g. inference time reasoning models).
- Question about the contribution of the KVFundaBench
    - The benchmark appears to be a collection of existing datasets evaluated under the few-shot CoT prompting setting for KV cache compression. Framing this as measuring the “degradation of fundamental capability” is misleading; the main contribution seems to be evaluating KV cache compression methods in few-shot CoT contexts, which does not generalize to other scenarios. Furthermore, Section 3.3 and Figure 5 do not specify how many few-shot examples are used for each benchmark.
- Concerns about the effectiveness of ShotKV
    - If the focus is solely on few-shot CoT prompting as a testbed for KV cache compression, the paper should include a simple baseline: reducing the number of shots at random (instead of selecting shots based on attention scores, as ShotKV does). Comparing ShotKV against this random-shot baseline would clarify its actual benefit.
- Missing results on real latency improvements
    - The paper does not report end-to-end latency reductions. It would be important to show how much latency is reduced in the prefill and decoding steps compared to FullKV.

**Questions:**

See the above weakness section.

**Details Of Ethics Concerns:**

I do not have ethics concern regarding this paper.

---

> ### Author Response · Authors · 2025-11-25
> **Response to Reviewer yiKU (1/2)**
>
> ### Weakness 1: Clarity of Setup & Compression Timing We apologize for the lack of clarity.
>
> We sincerely apologize for the lack of clarity regarding the experimental setup and method details. We will optimize the description in the revised version.
> 1. Evaluation Setting: Our core experimental setting is indeed Few-shot Chain-of-Thought (CoT). The specific number of shots and hyperparameters for each benchmark are detailed in Appendix C.1 (Table 4). For instance, we used 8-shot for GSM8K and 5-shot for MMLU. We will move this table to the main text to improve readability.
> 2. Compression Timing: ShotKV employs a "Prefill-Decoding Separation" strategy.
>     - Prefill Phase: Compression occurs only once. After processing the prompt, we calculate the importance score of each shot and retain key shots based on the prefill compression ratio. This part of the KV cache is then fixed and does not change during generation.
>     - Decoding Phase: Compression is performed dynamically. For newly generated tokens, we use a dynamic strategy (similar to Top-K selection) to maintain the KV cache based on attention scores at each step. The detailed algorithm is provided in Appendix D.1 (Algorithm 1).
>
> ### Weakness 2: Questions about generalizability & effectiveness of the method (WK/CSR comparable, specialized for few-shot)
> 1. Regarding Effectiveness: The primary goal of ShotKV is not to surpass SOTA on all tasks, but to address the performance collapse in "highly sensitive tasks" (such as Arithmetic Reasoning and Long-Context Generation).
>     - WK/CSR Tasks: As stated in Observation 1, these tasks are inherently robust to compression, and most methods (including StreamingLLM) maintain performance comparable to FullKV. Therefore, ShotKV's comparable performance here is expected.
>     - AR and Long-Gen Tasks: These are the critical pain points. In AR tasks (Figure 5c), ShotKV demonstrates superior stability compared to ChunkKV and PyramidKV when the compression ratio is below 20%. In the Long-Context Generation task (LG-GSM8K) shown in Table 2, ShotKV achieves an accuracy of 47.33% at a 40% compression ratio, significantly outperforming StreamingLLM (39.50%) and H2O (32.66%).
> 2. Regarding Generalizability: The core concept of ShotKV is preserving "semantic units" rather than just "shots." We provided a generalization experiment on a non-few-shot task (HotpotQA, Document QA) in the Table 6. By treating sentences/paragraphs as semantic units for compression, Table 10 shows that ShotKV achieves a score of 43.60 at a 10% compression ratio, outperforming H2O (40.84) and StreamingLLM (40.27). This proves the method effectively generalizes to non-few-shot scenarios.
>
> **Table: LLaMa-3-8B-Instruct Performance on HotpotQA (10% Compression Ratio).** Demonstrating generalization to non-few-shot tasks.
>
> | Method | Score |
> | :--- | :---: |
> | FullKV | 45.55 |
> | StreamingLLM | 40.27 |
> | H2O | 40.84 |
> | SnapKV | 43.36 |
> | PyramidKV | **43.80** |
> | ChunkKV | 43.27 |
> | **ShotKV (Ours)** | **43.60** |
>
> ### Weakness 3: Question about the contribution of the KVFundaBench
> 1. Contribution: The core contribution of KVFundaBench lies in its novel "evaluation perspective." Existing benchmarks like LongBench or NIAH primarily focus on "retrieval" capabilities, whereas we aim to reveal the potential degradation of "Fundamental Abilities" under compression. Our experiments uncovered that Arithmetic Reasoning (AR) and Long-Context Generation are extremely sensitive to compression (Observations 1 & 6), while WK/CSR are robust. These insights regarding the "divergence in capability degradation" are not covered by existing benchmarks.
>
> ### Weakness 4: Concerns about the effectiveness of ShotKV
> We thank the reviewer for this insightful suggestion. We would like to clarify that ShotKV operates specifically at the KV cache level, compressing the internal memory representations while maintaining the full semantic structure of the prompt. This is distinct from token-level compression (i.e., reducing the number of shots directly).
>
> Simply reducing the number of shots (token-level pruning) often leads to a significant degradation in performance, as it fundamentally removes the demonstrations required for in-context learning. In contrast, ShotKV aims to retain the benefits of multiple shots while reducing memory footprint. We agree that comparing against a "random shot reduction" baseline would valuable to quantify this advantage. We will include this comparison in the final version to empirically demonstrate the superiority of KV-level compression over simple token removal.

---

> > ### Author Response · Authors · 2025-11-25
> > **Response to Reviewer yiKU (2/2)**
> >
> > ### Weakness 5: Missing results on real latency improvements
> > We agree that efficiency metrics are crucial.
> > - As detailed in Table 5, in a long-context generation setting with an input length of 8192 and output length of 4096, ShotKV reduces latency by 11.3% and improves throughput by 13.1% compared to FullKV.
> > - These results confirm that ShotKV delivers tangible inference acceleration while maintaining high performance. We will move these critical data points from the appendix to the main text to address your concern.
> >
> > **Latency and throughput comparison between ShotKV and FullKV (Input 8192 / Output 4096).** Experiments performed on A40 GPU with batch size 1.
> >
> > | Method | Input Length | Output Length | Latency (s) $\downarrow$ | Throughput (T/S) $\uparrow$ |
> > | :--- | :---: | :---: | :---: | :---: |
> > | FullKV | 4096 | 4096 | 175.50 | 37.73 |
> > | **ShotKV** | 4096 | 4096 | **162.85 (+7.2%)** | **41.12 (+9.0%)** |
> > | FullKV | 8192 | 4096 | 183.42 | 55.93 |
> > | **ShotKV** | 8192 | 4096 | **162.78 (+11.3%)** | **63.24 (+13.1%)** |

---

### Author Response · Authors · 2025-11-25
**General Response to All Reviewers**

# General Response to All Reviewers

We thank all reviewers for their constructive feedback and the time dedicated to reviewing our work. We are encouraged that Reviewer Cx1q recognizes KVFundaBench as a "valuable structured evaluation suite" and ShotKV as "promising."

Based on the insightful comments, we have updated our manuscript and conducted additional experiments. We summarize the key clarifications and revisions below:

**1. Clarifying the "Long-Context" Scope (Response to Reviewer Yk6q & di7N)**
We clarify that our focus is on **"High-Density Reasoning"** rather than just input length. While traditional benchmarks (e.g., RULER, NIAH) focus on sparse retrieval over ultra-long contexts, our **Many-shot (50-shot) and Long-Generation** settings test the model's ability to maintain **semantic integrity** and **logical consistency** over thousands of tokens. Our results show that this "Dense Reasoning" context is far more sensitive to compression than "Needle-in-a-Haystack" retrieval, revealing a critical blind spot in existing evaluations.

**2. New Baselines & Generalization (Response to Reviewer yiKU & Cx1q)**
* **Generalization to Non-ICL Tasks:** We added experiments on **HotpotQA** (Document QA). By treating sentences/paragraphs as semantic units, ShotKV outperforms H2O and StreamingLLM, demonstrating its versatility beyond few-shot scenarios.

**3. Efficiency Metrics (Response to Reviewer Cx1q & Yk6q)**
We have added concrete efficiency data (Input 8192 / Output 4096) to the main text. Despite the overhead of score calculation, ShotKV achieves:
* **11.3% reduction in end-to-end Latency**
* **13.1% increase in Throughput**
compared to FullKV, confirming its practical viability for deployment.

**4. The Core Value of KVFundaBench**
We emphasize that the novelty of KVFundaBench lies in **systematizing the "Task-Dependent Degradation."** We are the first to empirically categorize tasks into "Compression-Robust" (Knowledge/Retrieval) and "Compression-Sensitive" (Reasoning/Long-Gen). This taxonomy explains why previous methods (like SnapKV) succeed on LongBench but fail on our rigorous reasoning tests, guiding the community toward safer compression strategies.

We hope these responses and new results address your concerns. We remain open to further discussions.

Best regards,
Authors

---

### Note · Authors · 2025-12-18

I have read and agree with the venue's withdrawal policy on behalf of myself and my co-authors.